# Mapping Seasonal Snow Melting in Karakoram Using SAR and Topographic Data

Shiyi Li[1], Lanqing Huang[1,3], Philipp Bernhard[1], and Irena Hajnsek[1,2]

[1]Chair of Earth Observation and Remote Sensing, ETH Zurich, Zurich, Switzerland
[2]Microwave and Radar Institute, German Aerospace Center (DLR), Wessling, Germany
[3]Center for Polar Observation and Modeling, University College London, London, United Kingdom

**Correspondence:** Shiyi Li (shiyi.li@ifu.baug.ethz.ch)

**Abstract.** Mapping seasonal snow melting is crucial for assessing its impacts on water resources, natural hazards, and regional climate in Karakoram. However, complex terrain in the high mountain region posed great challenges to remote sensing based wet snow mapping methods. In this study, we developed a novel framework that incorporated Synthetic Aperture Radar (SAR) and topographic data for robust and accurate mapping of wet snow over the Karakoram. Our method adopted the Gaussian Mixture Model (GMM) to adaptively determine a Wet Snow Index (WSI), and computed a Topographic Snow Index (TSI) considering the impact of terrain on wet snow distribution to improve the accuracy of mapping. We validated the mapping results against Sentinel-2 snow cover maps, which demonstrated significantly improved accuracy using the proposed method. Applied across three major water basins in Karakoram, our method generated large-scale wet snow maps and provided valuable insights into the temporal dynamics of regional snow melting extent and duration. This study offers a practical and robust method for snow melting monitoring over challenging terrains. It can contribute to a significant step forward in better managing water resources under climate change in vulnerable regions.

## 1 Introduction

Monitoring seasonal snow melting is of global importance within cryosphere studies, given the profound and far-reaching impacts of snow on climate, hydrology, and ecosystems. Snow cover plays a crucial role in modulating the Earth's energy balance by altering surface albedo, thereby exerting cooling effects on the terrestrial surface and influencing climate patterns at local and regional scales. Notably, in high-altitude areas, the accumulation of snow serves as a primary water source for downstream flows and governs the runoff dynamics in many mountainous basins (Barnett et al., 2005).

The Karakoram region, characterized by its elevated topography and unique climatic conditions, is of exceptional significance in snow cover monitoring. Situated at the center of the Hindukush-Karakoram-Himalaya (HKH) mountain system, the Karakoram is renowned for hosting some of the world's highest peaks and harboring the largest alpine glacier system outside the polar regions (Nie et al., 2021). Across its expansive landscape, snow and ice reserves are substantial, encompassing an area exceeding 20,000 km$^2$, with seasonal snow covering nearly 90% of this expanse (Lund et al., 2020; Hasson et al., 2014; Xie et al., 2023).

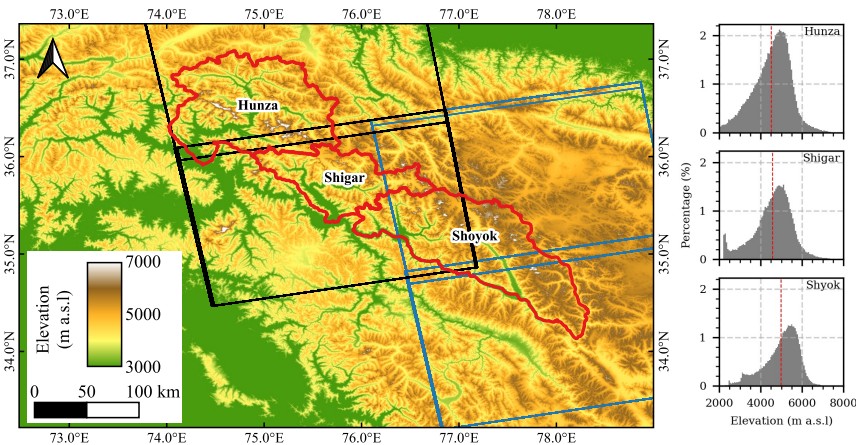

**Figure 1.** Geolocation of the Karakoram region overlaid on the COP-30 DEM. The study regions includes three major water basins: Hunza, Shigar, and Shyok, which are delineated in red. Elevation histograms of the tree basins are shown on the right panel. Median elevation of basins are indicated with red vertical lines in histograms. Footprint of S1 images used in the study are show with black and blue boxes (black: relative orbit 27; blue: relative orbit 129).

In the expansive and challenging terrain of the Karakoram region, ground-based observation methods struggle to effectively
cover the vast and rugged terrain, and hence remote sensing techniques have become the primary approach for snow cover
mapping. While space-borne optical and multi-spectral sensors like MODIS, Landsat-7/8, and Sentinel-2 (S2) have been
employed in numerous studies, their reliability is often compromised by cloud cover and illumination conditions (Immerzeel
et al., 2009; Hasson et al., 2014; Tahir et al., 2011, 2015; Khan et al., 2015; Wu et al., 2021). To overcome this limitation,
Synthetic Aperture Radar (SAR) presents a valuable alternative for monitoring snow regardless of weather and daylight
conditions.

The foundation of snow classification in SAR imagery was established based on the unique backscattering responses
generated from the distinctive interactions of snow with SAR signals. For dry snow, radar signals can penetrate through the
snowpack down to a specific depth depending on the signal wavelength, and thus generates strong backscattering signal at
the snow-ground boundary (Baghdadi et al., 1997; Millan et al., 2015). As the snowpack undergoes melting, its liquid water
content increases in the wet snow pack and causes high dielectric losses, resulting in significant reductions in backscattering
intensities (Baghdadi et al., 1997; Nagler et al., 2016). Based on the change of backscattering intensities, early wet snow
detection methods were developed using the ratio of SAR backscattering coefficients (Baghdadi et al., 1997; Nagler and Rott,
2000). The ratio was derived from SAR images acquired during the snow melting season and reference images obtained under
snow-free or dry-snow conditions. An empirical threshold of -2 dB on C-Band Sentinel-1 (S1) ratio images was proposed to
classify snow, distinguishing wet from dry snow and had been proven to be effective (Nagler et al., 2016).

Subsequent refinements were proposed to enhance the algorithm for robust application on various ground surface types. For example, sigmoid functions were introduced as a soft threshold to replace the -2 dB, and it was shown to be effective in resolving the uncertainties arising from mixed pixels of wet snow and other constituents (Malnes and Guneriussen, 2002; Longepe et al., 2009; Rondeau-Genesse et al., 2016). Various strategies for selecting bias-free reference images were devised, such as choosing a specific reference scene during winter (Koskinen et al., 1997), averaging multiple images over a defined period (Luojus et al., 2006), or employing linear interpolation between images at the beginning and end of the melting period (Pettinato et al., 2014). In practice, auxiliary data were usually combined to improve the accuracy of snow detection, such as Digital Elevation Models (DEMs), land category maps, meteorological model, and snow cover maps derived from optical multi-spectral sensors (Nagler and Rott, 2000; Liu et al., 2022b, a; Nagler et al., 2016; Tsai et al., 2019). Recent developments in machine learning also brought opportunities to further improve SAR snow mapping. Supervised classification algorithms, such as support vector machine and random forest, were applied to different SAR products and have shown promising results (Tsai et al., 2019; Huang et al., 2011). Deep learning algorithms were also exploited and have exhibited great potential in achieving more robust and accurate wet snow classification (Liu et al., 2023).

Despite the efforts of improving the method for robust SAR-based snow mapping, challenges remain in mountainous regions such as the Karakoram, where complex topography may strongly distort SAR signals and thus lead to considerable uncertainties when applying the single-valued thresholds for snow classification (Snapir et al., 2019; Lund et al., 2020; Karbou et al., 2021b, a). Furthermore, large-scale application of the wet snow mapping in Karakoram requires a method to be adaptively responsive to basin-specific variations, posing practical challenges to efficient method development (Rondeau-Genesse et al., 2016). ML techniques may offer versatile solutions, but their application in this region is limited by the availability of training data (Tsai et al., 2019; Liu et al., 2023).

To address these challenges, this study proposed a novel framework that integrated SAR and topographic data for versatile and robust wet snow mapping covering three major water basins of the Karakoram. In the first step, we employed an unsupervised learning algorithm, namely the Gaussian Mixture Model (GMM), to adaptively determine the Wet Snow Index (WSI). Secondly, a Topographic Snow Index (TSI) was calculated to account for the influence of topography on snow distribution. We applied the proposed method to a time-series of SAR imagery acquired by Sentinel-1 (S1) between 2017-2021, and generated regional-wide wet snow maps of high spatial-temporal resolution. The validation using S2 images showed considerable improvements comparing to conventional threshold-based methods. Crucial snow dynamic variables including the Wet Snow Extent (WSE) and Snow Melting Duration (SMD) was derived from the snow maps, demonstrating the significance of closely monitoring wet snow in watershed management.

The paper is organized as the following. Section 2 introduces the study site and data. Section 3 explains the proposed method in detail. Section 4 presents the result of the study, including the validation and the snow dynamic variables. Section 5 discusses the method and its implications for snow mapping. Section 6 concludes the study and provides an outlook for the future development.

**Table 1.** Information on Sentinel-1 (S1) images used for the generation of wet snow maps. Ascending pass cross over the study region in the late afternoon (around 18:00 local time).

|  | Relative Orbit | Orbit Direction | Revisit Interval | Start Date | End Date | Number of Acquisitions |
|---|---|---|---|---|---|---|
| Hunza | 27 | Ascending | 12 days | 2017-02-05 | 2021-12-29 | 148 |
| Shigar | 27 | Ascending | 12 days | 2017-02-05 | 2021-12-29 | 148 |
| Shyok | 129 | Ascending | 12 days | 2017-03-20 | 2021-12-24 | 146 |

## 2 Study area and data

### 2.1 Study Area

The Karakoram region, spanning extensively across parts of Pakistan, India, and China, is bordered by some of the highest mountain systems on Earth, including the Himalayas, the Pamir Mountains, and the Hindu Kush Mountain Ranges (Figure 1). The study area encompasses the majority of the Karakoram, covering an expansive geographical domain of approximately 450,000 km$^2$. This region extends from approximately 35°N to 38°N latitude and 76°E to 78°E longitude, characterized by a wide range of altitudes from around 2,000 m a.s.l (above sea level) in the valleys to well over 8,000 m a.s.l at the highest summits. The extreme topographic variation gives rise to rugged terrain, including steep valleys and towering mountain peaks.

The Karakoram is situated upstream of both the Upper Indus Basin and the Tarim River Basin. It covers three significant watersheds: Hunza, Shisgar, and Shyok, as shown in Figure 1. These basins serve as the upstream sources of the UIB, and contributes 65-85% of annual flows to the Indus River with the melting water from snow and glaciers, sustaining the livelihoods of millions of residents residing within these basins (Archer, 2003; Immerzeel et al., 2009; Forsythe et al., 2012). The hydrological importance of Karakoram emphasized the important role of mapping snow melting in the region.

### 2.2 Data

#### 2.2.1 Sentinel-1 SAR Imagery

The SAR imagery used in our research was acquired by the space-borne C-band S1 SAR sensor. The S1 SAR satellite provides high geolocation accuracy and a short orbit repeat cycle of 12 days, facilitating precise and frequent monitoring of snow melting (Nagler et al., 2016).

In this study, we use single-look-complex (SLC) data from S1 acquired in Interferometric Wide (IW) swath mode. These data cover scenes with a swath width of $250\,\mathrm{km}$ at a spatial resolution of $\sim 5 \times 20\,\mathrm{m}$ in range and azimuth direction. Both VV and VH dual-polarization data were employed for the following analysis. The details of the S1 images utilized in this study, including orbit numbers and acquisition dates, are provided in Table 1.

### 2.2.2 Copernicus Digital Elevation Model

Digital Elevation Models (DEM) provide essential topographic information crucial for both SAR image preprocessing and snow distribution mapping, particularly in the rugged landscapes of mountainous regions. In this study, we employed the Copernicus Global 1-arcsec (COP-30) DEM, recently released by the European Space Agency (ESA) in 2020, to facilitate the SAR image processing and snow mapping. The COP-30 DEM product was derived from the TanDEM-X SAR data acquired between 2010 and 2015, providing global coverage at a resolution of 30 meters and a vertical root mean square error as low as $1.68\,\text{m}$ (Li et al., 2022; Guth and Geoffroy, 2021).

The COP-30 DEM data covering the study regions was downloaded through the Copernicus Space Component Data Access PANDA Catalogue (European Space Agency and Airbus, 2022), as shown in the base map in Figure 1. The DEM products are referenced in Geographic Coordinates using the World Geodetic System 1984 (WGS84). The vertical heights are reference to the EGM2008 geoid model.

### 2.2.3 Sentinel-2 Level-2A Imagery

To validate the snow maps generated using the proposed method, we also derived snow cover maps on selected dates using multi-spectral S2 images. The S2 sensor operates in a sun-synchronous orbit with a revisit time of 5 days (Drusch et al., 2012). Equipped with 13 spectral bands ranging from visible, near-infrared to shortwave infrared, S2 images offer valuable spectral information for land cover characterization and has been widely used in snow cover mapping. In this study, we used the S2 Level-2A (L2A) products to generate snow cover maps for validation purposes. The L2A product is orthorectified Bottom-Of-Atmosphere surface reflectance data, that are derived through the atmospheric correction of the Level-1C products using the Sen2Cor method (Main-Knorn et al., 2017). Practically useful supplementary data is also included in the L2A product, including the cloud and snow confidence maps and the scene classification map that identifies elements like clouds, cloud shadows and snow. The spatial resolution of images under different spectral bands varies from 10 to 60 meters.

We selected S2 L2A images taken during the summer months with minimal cloud cover as potential candidates. From these, only images that could be matched with S1 SAR image within a window of $\pm 3$ days were used to generate reference snow maps with the Let-it-snow (LIS) algorithm (Section 3.5; (Gascoin et al., 2019)). The LIS algorithm employed RGB spectral bands B2 (blue), B3 (green), and B4 (red), as well as infrared bands B8 (NIR) and B11 (SWIR) for snow-cloud-ice classification. We accessed the S2 data through the Copernicus Open Access Hub.

### 3 Methodology

This section describes the proposed method for integrating SAR and topographic data to map the melting wet snow. The key steps of the method are summarized in Figure 2.

The raw S1 SLC data were first pre-processed to generate backscatter images. Same-orbit SLC time-series were co-registered to a common reference scene to accurately align the geo-location of pixels. Then we multi-looked each SLC with a window

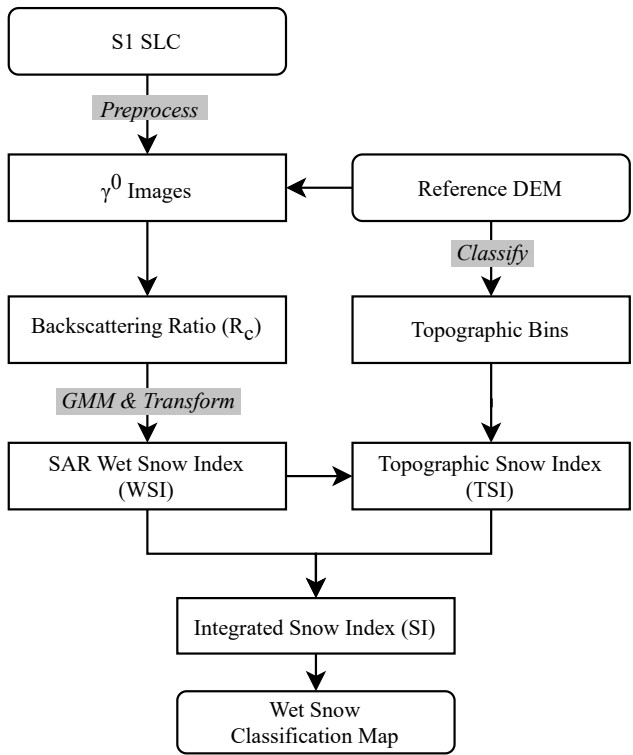

**Figure 2.** The developed method for mapping snow melting. Critical processing steps are shown in the gray box. The COP-30 DEM was used as the Reference DEM. GMM stands for the Gaussian Mixture Model.

size of $12 \times 1$ (rg $\times$ az) to obtain backscattering intensity images with squared pixel spacing of approximately $14 \times 14$ m. The intensity images were further converted to $\gamma^0$ images with terrain-based radiometric correction (Frey et al., 2013). All preprocessing steps were conducted using the GAMMA software (Werner et al., 2000).

### 3.1 SAR Backscattering Ratio

With the preprocessed $\gamma^0$ images, we derived the composite backscatter ratio ($R_c$) following the method proposed by Nagler et al. (2016). The $R_c$ metric combines the backscatter ratios from both the VV and VH channels to comprehensively assess surface condition changes associated with snow melting. This approach incorporates a weighting factor $W$, which is determined by the local incidence angle (LIA, $\theta$), to account for variations in backscatter due to differing incidence angles. This adjustment enables more robust snow melt detection across varying terrain geometries.

To compute $R_c$, we first calculated the SAR backscatter ratio $R_i$ for each polarization $i \in$ vv, vh using the following equation:

$$R_i = \gamma_i^0 / \gamma_{i,ref}^0 \tag{1}$$

where $\gamma^0_{i,ref}$ represented a reference image constructed using the average of multi-year winter images. Note that this differs from other alpine regions, such as the Alps, where summer months are often used as the reference due to snow-free conditions. In the Karakoram, due to the all year long snow cover at the higher elevations, we used winter images as reference to leverage the dry snow conditions in winter for highlighting the contrast in backscattering intensity between summer wet snow (lower intensity due to water absorption) and winter dry snow

With $R_i$ for each polarization, the composite ratio $R_c$ was calculated as a weighted sum of $R_{vv}$ and $R_{vh}$

$$R_c = W R_{vv} + (1 - W) R_{vh} \tag{2}$$

where the weighting factor $W$ is defined based on the LIA as:

$$W = \begin{cases} 0 & (\theta < \theta_1) \\ 0.5(1 + \frac{\theta - \theta_1}{\theta_2 - \theta_1}) & (\theta_1 \le \theta \le \theta_2) \\ 0.5 & (\theta > \theta_2) \end{cases} \tag{3}$$

with $\theta_1 = 20°$ and $\theta_2 = 45°$.

## 3.2 Wet Snow Index

While $R_c$ alone effectively indicates surface condition changes, it can be sensitive to local variations and does not inherently incorporate adaptive boundaries for wet snow classification. Therefore, instead of directly applying a threshold to $R_c$ for wet snow classification, we propose using a GMM to convert $R_c$ into a Wet Snow Index (WSI) to have a probabilistic measure that better captures the varying conditions of wet snow across different terrains. By leveraging the density distribution of $R_c$ values, the WSI enables a dynamic scaling of the classification based on the underlying distribution of $R_c$ values.

The GMM is an unsupervised probabilistic model for clustering and density estimation. We configured the GMM to identify two clusters corresponding to the wet snow pixels and the no snow (or dry snow) pixels as

$$P(R_c) = \sum_{i=1}^{2} \pi_i \cdot N(R_c | \mu_i, \sigma_i), \tag{4}$$

where $P(\cdot)$ the probability density function (PDF) of $R_c$, $\pi_i$ the $i$-th Gaussian component's mixing coefficient for wet snow ($i = 1$) and no (or dry) snow ($i = 2$) clusters, and $N(R_c | \mu_i, \sigma_i)$ the Gaussian distribution with mean $\mu_i$ and standard deviation $\sigma_i$. We used the full covariance structure in the GMM model, i.e. each component has its own general covariance matrix, after testing different types of covariance structures.

To train the GMM and determine its parameters ($\pi_i$, $\mu_i$, and $\sigma_i$), we randomly sampled $10^6$ unlabeled pixels from the summer $R_c$ images of each basin, and employed the expectation-maximization (EM) algorithm to fit the model for each basin with the sampled pixels (Dempster et al., 1977). During the training, the maximum number EM iterations were set to 100, and the convergence threshold was set to $10^{-3}$. With the estimated GMM parameters, the WSI can be determined using a modified logistic function:

$$WSI = \frac{L}{1 + e^{k(x - x_0)}} \tag{5}$$

with

$$\begin{cases} k = \dfrac{|\mu_1 - \mu_2|}{\sigma_1 + \sigma_2} \\ x_0 = \dfrac{\mu_1 + \mu_2}{2} \end{cases}$$

where $k$ the slope factor, $x_0$ the logistic curve's midpoint and $L$ the carrying capacity representing the supremum of the function. Here the carry capacity $L$ was set to 10 to amplify the differences between pixels of different $R_c$ values.

In the WSI logistic function, the slope factor $k$ is determined by the separation between the two clusters, providing a flexible and adaptive control over the sensitivity of WSI to the difference between snow conditions. In the case when the two clusters are perfectly distinct from each other (i.e. $|\mu_1 - \mu_2| \gg \sigma_1 + \sigma_2$), the WSI would be transformed into a step function and thus

effectively act as a single-value hard-threshold at $x_0$. Therefore, the Nagler's method can be taken as a special case under this assumption with $x_0 = -2$ dB. Contrarily, the mixed clusters (i.e. $|\mu_1 - \mu_2| \ll \sigma_1 + \sigma_2$) would lead to progressively flattened WSI and soft segmentation boundaries. A similar form of logistic function was proposed by Rondeau-Genesse et al. (2016), which was determined with empirical parameters and was used as soft thresholds to replace the hard-threshold of -2 dB on $R_c$. In our approach, the GMM allows for an adaptive choice of the parameter $k$ based on the distribution density of $R_c$, thereby

enabling flexible and robust applications in large scale.

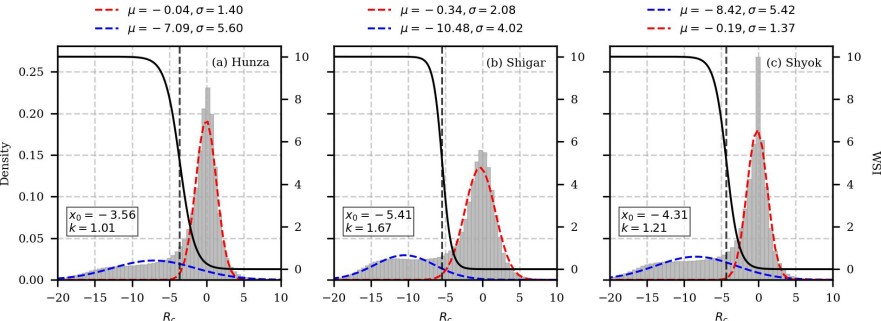

**Figure 3.** Parameters of the GMM and WSI used for (a) Hunza, (b) Shigar and (c) Shyok. Gray shaded histogram shows the density of $R_c$ of the sampled points. Red and blue dashed curve represent the PDF (scaled on the left y-axis) of the two clusters in the GMM. Black solid lines (scaled on the right y-axis) are the WSI. The vertical black dashed line marked the center of WSI curves (where $R_c = x_0$). The mean ($\mu$) and standard deviation ($\sigma$) of GMM are reported above the panel.

### 3.3 Topographic Snow Index

Given the strong impact of terrain properties on snow distribution, we introduce the TSI as a component of the proposed method to account for the terrain influence on snow presence.

Terrain properties, including the elevation, slope, and aspect, collectively influence the spatial and temporal distribution of

snow. Comparing to lower altitudes, regions of higher elevation experience lower temperatures and are conducive to more

snow accumulation. The steepness of slopes and the orientation of aspects, on the other hand, impact the snow distribution through the solar illumination and wind exposure. To take these factors into account, we caluculated the TSI in two steps. First, we derived the discrete topographic bin maps using the COP-30 DEM by partitioning the terrain with its elevation, slope and aspect. The partition was based on slope below or above $20°$, elevation in every 100m, and aspect in every $15°$. The topographic bin map can effectively capture the localized terrain attributes that influence the occurrence of snow. The median WSI value within each topographic bin was then calculated to obtain the TSI, so that the regional propensity of wet snow occurrence under the specific topographic conditions could be quantified.

Example TSI distribution for Hunza at different elevation, slope and aspect are presented in Figure 4.

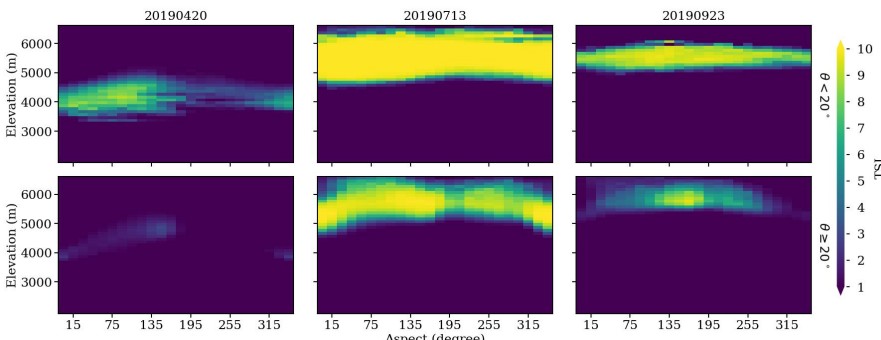

**Figure 4.** TSI values at different elevation (y-axis), aspect (x-axis) and slope classes (top row: flat slope with $\theta < 20°$, bottom row: step slope with $\theta < 20°$) for Hunza basin. Three observation dates in spring (left column), summer (middle column) and late autumn (right column) are displayed. Aspect of 0, 90, 180 and 270 degrees align with the north, east, south and west direction, respectively.

## 3.4 Integrated Snow Index

As illustrated in Figure 2, the final step generated an integrated Snow Index (SI) map by performing pixel-wise multiplication of WSI and TSI. This multiplication scales the WSI by incorporating terrain characteristics, thereby linking the observed SAR backscattering ratio directly with terrain properties.

In order to classify the integrated SI into binary snow maps, it is crucial to apply an adaptive threshold that accounts for the variation in topographic features across different basins. The variation in SAR backscatter response within a basin is inherently handled by the GMM when deriving the WSI. In contrast, the TSI is time-varying and basin-specific, requiring an optimal coefficient to condition the SI for classification. To determine this coefficient, we performed a sensitivity analysis, evaluating F1-score, precision, and recall across different values using the S2 validation snow map. The results (figure A1) demonstrate that Hunza and Shyok exhibit similar responses, with optimal coefficients close to 3.5, while Shigar reaches its optimum at approximately 2.5. However, to avoid overfitting to specific basins or validation dates, we selected 3.5 as an overall coefficient to balance classification performance across all basins. This coefficient also reflects a moderate threshold applied to the TSI to determine the overall SI threshold for each basin.

The threshold was then calculated using the following equation:

$$\text{SI Threshold} = 3.5 \times \text{WSI}|_{R_c = -2} \tag{6}$$

where $\text{WSI}|_{R_c = -2}$ represented the WSI at a backscatter ratio $R_c = -2 \; dB$ for each basin. This value is basin-specific, allowing the threshold to adapt based on each basin's distinct characteristics. Together, these conditions form an integrated, basin-adaptive thresholding mechanism, combining SAR backscatter and topographic information into a single index to determine the SI threshold.

Note that while the SI threshold is basin-specific, it is time-independent. The WSI is derived from a GMM trained on samples collected from multiple summer scenes over several years, ensuring that it represents an aggregated measure for each basin and is not tied to individual scenes or seasons. This design ensures robustness to seasonal variations in liquid water content and enables consistent application across different validation dates.

## 3.5  Sentinel-2 Snow Cover Maps

The proposed method was validated using S2 snow cover maps generated following the LIS algorithm proposed by Gascoin et al. (2019). Before running the LIS algorithm, the input S2 multi-spectral bands were resampled to a pixel size of $20\text{m} \times 20\text{m}$ to match the resolution of different bands. The COP-30 DEM was also resampled to the same pixel size as the S2 images.

The LIS algorithm started with generating provisional snow masks by applying thresholds on the Normalized Difference Snow Index (NDSI) and the red band reflectance ($\rho_{\text{red}}$) with the condition:

$$(NDSI > n_i) \;\; AND \;\; (\rho_{\text{red}} > r_i) \tag{7}$$

where $n_i = 0.4$ and $r_i = 0.2$ (Gascoin et al., 2019). This step was designed to identify snow-covered areas while excluding non-snow surfaces such as vegetation and bare ground. However, this approach could sometimes exclude snow-covered pixels due to errors in cloud masking. To correct the errors, a refinement step was introduced to reassign dark cloud pixels that were initially misclassified. Following Gascoin et al. (2019), dark clouds were identified by applying a threshold of 0.3 on the bi-linearly down-sampled red band, which reduced the resolution of the red band from 20m to 240m by a factor of 12. This process helped to exclude cloud shadows and high-altitude cirrus clouds from the snow classification. Afterwards, the provisional snow masks were further refined using the basin snowline calculated from the COP-30 DEM. We calculated the total snow cover fraction (SCF) within every 100m elevation band using the provisional snow mask, and defined the snowline using the lowest elevation band where the SCF exceeded 30%. For pixels identified as dark clouds above the determined snowline, the conditions used in Equation (7) were reapplied with adjusted thresholds to account for the unique conditions at high altitudes. Specifically, the relaxed thresholds of $n_i = 0.15$ and $r_i = 0.04$ were used to classify snow pixels, and dark cloud pixels with $\rho_{\text{red}} > 0.1$ were reassigned as cloud, while other pixels were categorized as "no-snow." These adjusted thresholds help to differentiate snow from dark clouds in challenging high-altitude environments, ensuring a more accurate classification. Following the adjustment of the snow mask, we extended the LIS algorithm by further applying a threshold on the NIR band with $\rho_{\text{NIR}} > 0.4$ to classify glacier ice and water bodies from snow (Paul et al., 2016).

**Table 2.** Sentinel-1 (S1), Sentinel-2 (S2) Data and SI thresholds used for the validation. Different acquisition dates and adaptive SI thresholds were used for basins.

|        | S1 Date    | S2 Dates                 | SI Threshold |
|--------|------------|--------------------------|--------------|
| Hunza  | 2020-07-31 | 2020-07-29<br>2020-07-31 | 14.62        |
| Shigar | 2019-08-06 | 2019-08-06               | 5.44         |
| Shyok  | 2019-07-08 | 2019-07-07<br>2019-07-09 | 12.33        |

## 4 Results

### 4.1 Validation of Snow Classification Maps

The validation of snow classification results was carried out for three basins on specifically suited summer dates (Table 2). These dates were chosen based on conditions of minimal cloud cover and the shortest possible intervals between acquisitions of S1 and S2 images. As S2 snow cover maps classifies snow (both dry and wet) rather than only wet snow, we have chosen only the summer dates as listed in Table 2 to ensure that the S2 snow cover maps were mostly covered by wet snow. For Hunza and Shyok, two S2 images with an acquisition interval of 2 days were combined to achieve a complete coverage of the basin, whereas same day acquisitions of S1 and S2 was found for Shigar in 2019.

In the three basins, adaptive SI thresholds were used to generate the snow classification maps. The threshold values for each basin are also reported in Table 2. These thresholds provided basin-adaptive and time-independent classification boundaries to distinguish wet snow and dry snow or snow-free pixels.

Figure 5 demonstrates maps of $R_c$ and SI, as well as the SI snow map and the S2 snow map over the three basins. Compared to the $R_c$ map, the SI map shows much clearer boundary that separates wet snow pixels from non-or-dry snow pixels. Over glacier surfaces and valley regions (shown with the zoom-in insets), $R_c$ falls in the value range of $-4$ to $0$ dB, making it sensitive to the choice of threshold values. The $R_c$ map over these regions present noisy patterns, likely due to the complex scattering mechanisms on glacier surfaces. Over glacier surface, especially in the ablation zone, radar backscatter responses are highly variable due to refreezing, supraglacial features (e.g., crevasses, suncups, debris cover) and the presence of wet debris or bare ice (Scher et al., 2021). These features contribute to significant spatial variability in backscatter within a single pixel, making it challenging to distinguish dry snow, wet snow, and ice with $R_c$ alone. Additionally, glacier movement between winter and summer scenes introduces further variability, compounding the uncertainty in detection.

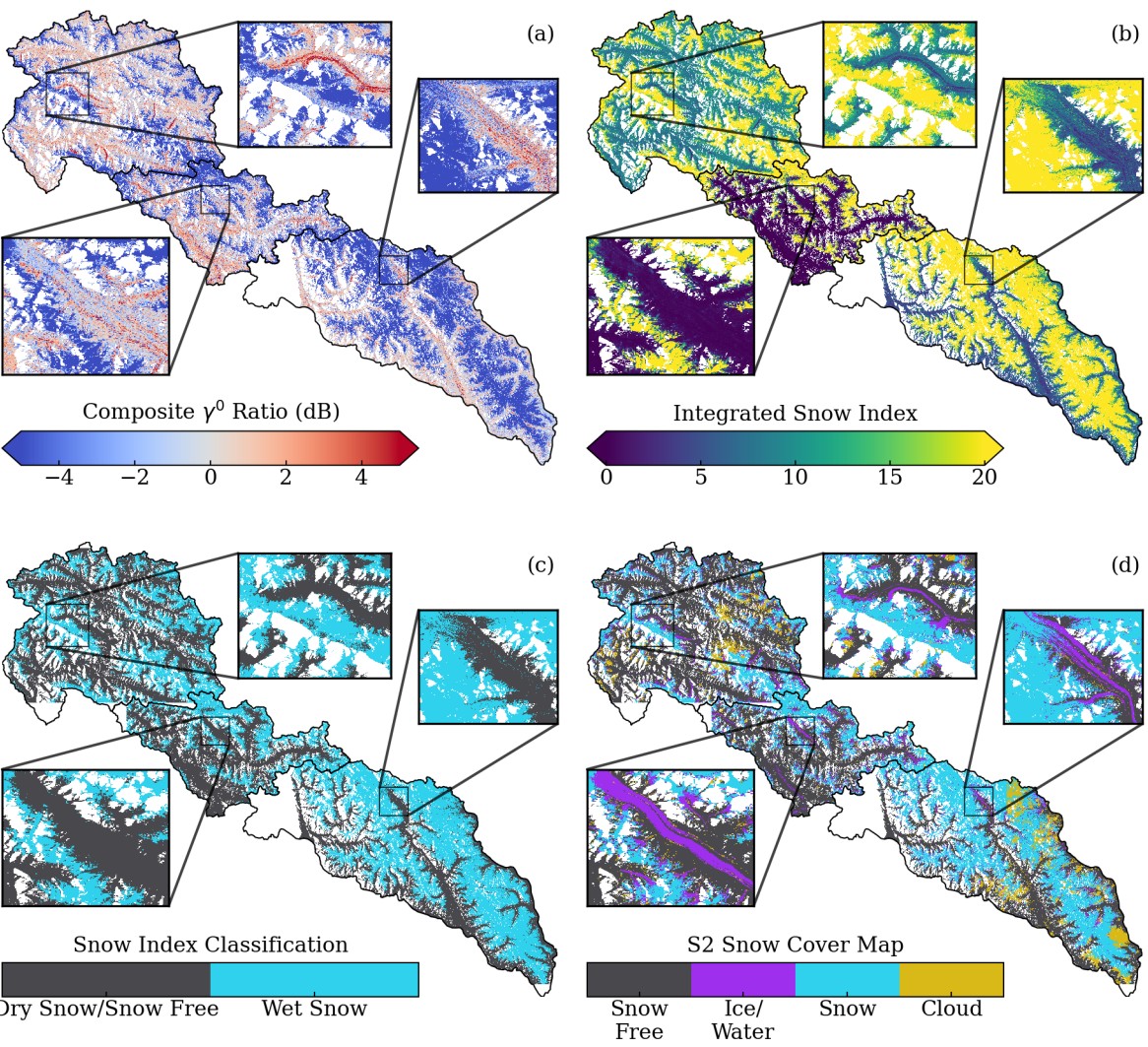

**Figure 5.** (a) $R_\mathrm{c}$, (b) Integrated SI, (c) SI Classification Map, and (d) S2 Snow Cover Map (as reference) for all three basins (Hunza, Shigar, and Shyok). The S2 snow map was generated using the Let-it-snow (LIT) algorithm, as described in Section 3.5. Zoomed insets provide a closer view of selected locations in each basin, highlighting performance differences on glacier surfaces.

In Figure 5(b), it is observed that the variation of SI values are within different range across the three basins. This is due to the different density distributions of $R_c$, WSI, TSI and SI in the three basins as shown in figure 6, underscoring the necessity of applying dynamically adaptive thresholds for each basin to generate classification results.

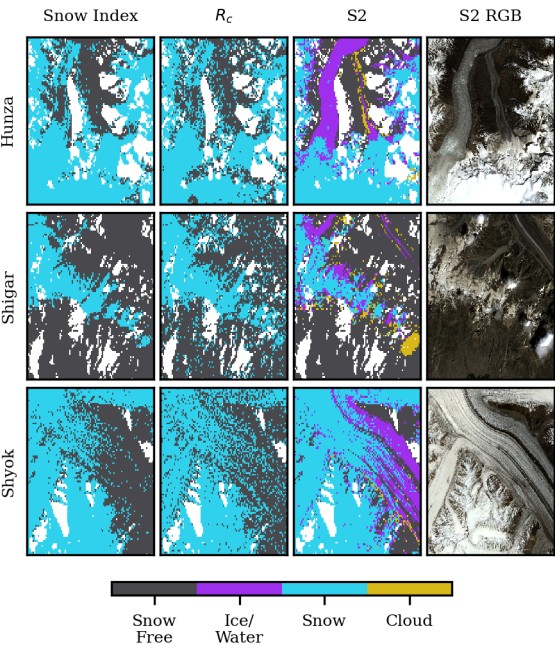

**Figure 6.** Detailed comparison between the snow classification map obtained using SI (left column), $R_c$ (middle column) and the S2 Snow Cover Map (right column, reference) for basin Hunza (top row), Shigar (middle row) and Shyok (bottom row). Class labels are indicated in the color bar. Note that the selected regions were located in the mid-elevation range of the area, hence the snow class in both S1 and S2 results both refereed to wet snow.

The snow maps in Figure 5(c) and (d) offer a visual comparison between the snow maps produced by the proposed method and the S2 LIS algorithm. A detailed comparison is further illustrated in Figure 7. Overall, the two methods show good agreement, demonstrating the effectiveness of the proposed approach. Notably, the SI-based classification exhibits reduced false positives and cleaner snow-free classifications in valleys and over glacier surfaces, as evident in the detailed comparison. This improvement is primarily attributed to the incorporation of topographic information through TSI, which smooths SI values within each topographic bin. However, a consistent mismatch in the ice/water category can be observed between the SAR (both SI and ratio methods) and S2 results, particularly over glacier surfaces. This discrepancy arises from the differing detection principles of the two approaches: the S2 results classify glacier ice using thresholds on the NIR band, while the SAR-based methods do not explicitly resolve glacier ice. On the observed date, glacier ice in the ablation zone may have partially melted, reducing the SAR backscatter ratio and leading to its misclassification as wet snow in the SAR results. As discussed earlier, glacier surfaces present unique challenges for SAR-based methods due to their complex scattering mechanisms. While the

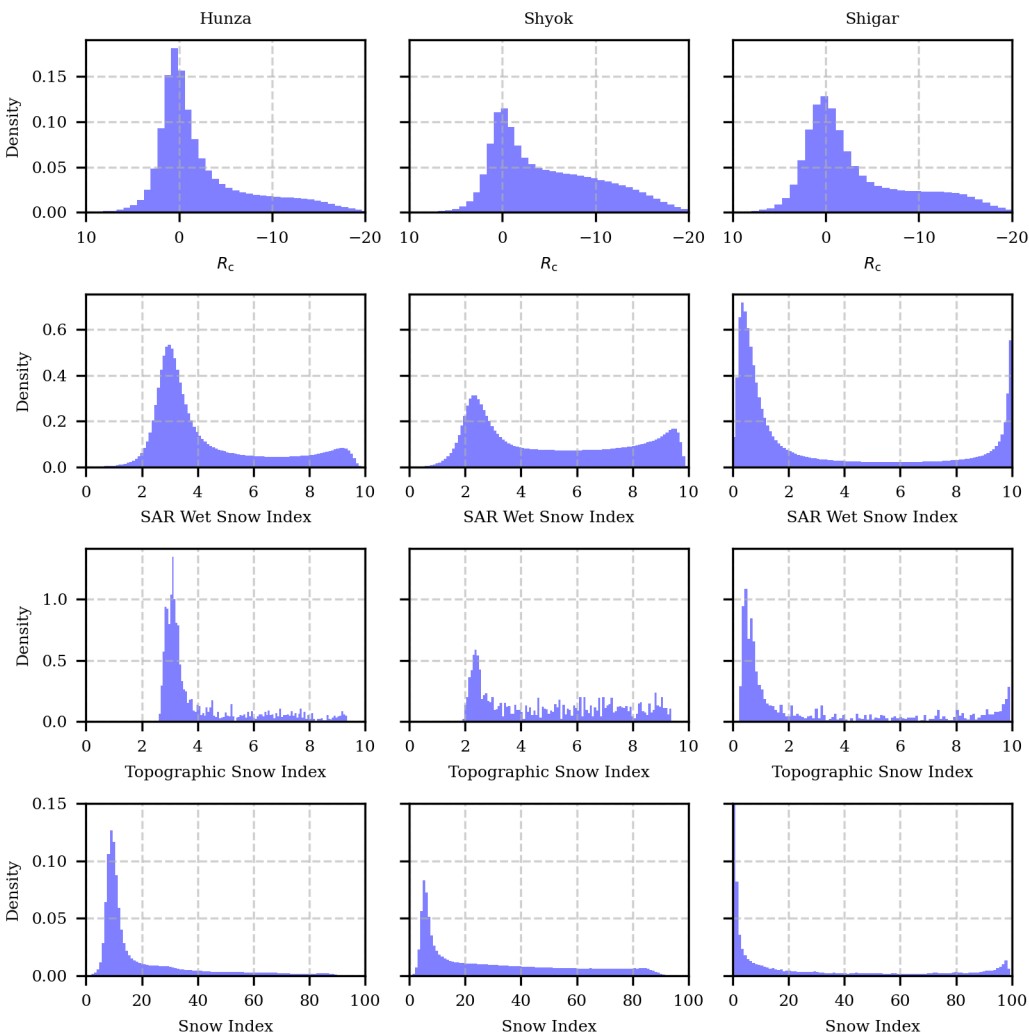

**Figure 7.** Density distribution of $R_c$, WSI, TSI and SI in three basins (left: Hunza, middle: Shyok, right: Shigar). The x-axis of $R_c$ is inverted as negative $R_c$ corresponds to higher probability of wet snow.

inclusion of TSI improves the robustness of our method by integrating topographic controls, it does not explicitly account for the heterogeneity of glacier surfaces. Consequently, glacier-specific conditions, such as localized melting or scattering from mixed ice-snow surfaces, may lead to underestimation or misclassification.

We further quantified the comparision using confusion matrix and the F1-score, as listed in Table 3. In the confusion matrix, S2 Snow Free (S2-SF) and S2 Snow (S2-S) pixels are used as negative and positive labels, respectively, while S1 snow classification maps designate no-or-dry snow (S1-N/D) and Wet Snow (S1-WS) pixels as their counterparts. To minimize errors caused by the presence of dry snow at high altitudes, areas above 5500 m a.s.l. were excluded from the calculation. This elevation threshold, representing 11.08% of pixels in Hunza, 12.95% in Shigar, and 30.43% in Shyok (19.60% in total), was chosen because such areas are consistently classified as non-melting zones due to persistently low air temperatures that inhibit snowmelt. Across all three basins, the proposed method shows an improvement in classification performance. In Hunza and Shyok, both true negative and true positive rates increased, while in Shigar, the true negative rate improved by 0.11, though the true positive rate decreased slightly by 0.04. Despite this, the overall F1-scores improved for all three basins, highlighting the method's enhanced precision and recall.

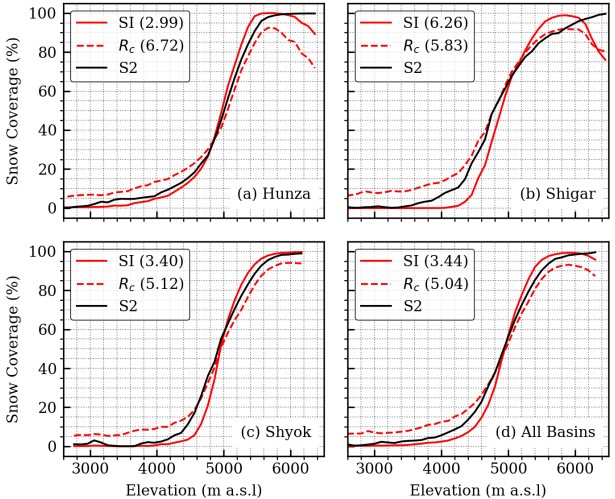

**Figure 8.** Snow coverage profile along elevation bands. Elevation bands were binned every 100m, and the snow coverage within a band was calculated based on the aggregated snow pixel percentage within every band. Numbers reported in legends measured the mean-absolute-error (MAE) between the profile curve of SAR wet snow classification and the S2 snow cover map.

We further evaluated the accuracy of the classification map using the elevation profile of snow distributions. As illustrated in Figure 8, the profile of snow coverage was analyzed along 100m elevation bands. Below 4500m a.s.l, snow coverage was overestimated by approximately 7% when using the $R_c$ thresholds, leading to a misrepresentation of the actual snow line, especially over the challenging conditions on glacier surfaces and valleys (Figure 5). In contrast, the SI method provided a snow classification closer to the S2 profiles at the low elevation regions. Between 4500m and 5500m a.s.l, the SI curve exhibited

**Table 3.** Confusion matrix and F$_1$ score between S1 snow classification maps and S2 snow cover maps. No-snow (S2-NS) and snow (S2-S) pixels in S2 snow cover maps are correspond to the negative and positive labels respectively. The associated labels in S1 snow classification maps are no-or-dry (S1-N/D) snow pixels and wet snow (S1-WS) pixels. Results of the proposed method are highlighted in bold, whereas the $R_c$ threshold based results are reported in normal font.

| Basin | | Confusion Matrix | | F1 Score |
| --- | --- | --- | --- | --- |
| | | S2-SF (N) | S2-S (P) | |
| Hunza | S1-N/D (N) | **0.93** \| 0.88 | **0.07** \| 0.12 | **0.86** \| 0.78 |
| | S1-WS (P) | **0.13** \| 0.21 | **0.87** \| 0.79 | |
| Shigar | S1-N/D (N) | **0.94** \| 0.83 | **0.06** \| 0.17 | **0.84** \| 0.79 |
| | S1-WS (P) | **0.22** \| 0.18 | **0.78** \| 0.82 | |
| Shyok | S1-N/D (N) | **0.90** \| 0.87 | **0.10** \| 0.13 | **0.93** \| 0.89 |
| | S1-WS (P) | **0.08** \| 0.13 | **0.92** \| 0.87 | |
| All Basins | S1-N/D (N) | **0.92** \| 0.86 | **0.08** \| 0.14 | **0.89** \| 0.84 |
| | S1-WS (P) | **0.12** \| 0.16 | **0.88** \| 0.84 | |

a noticeably steeper slope than the S2 curve, with an underestimation of snow coverage between 4000m and 5000m and an overestimation from 5000m to 5500m. This pattern suggests that the SI uncertainties in mixed snow conditions within transition zones might have led to a nonlinear exaggeration of the TSI response to snow cover. A more precisely calibrated TSI model could further align snow coverage profiles between the SI method and S2 results. Above 5500m a.s.l, where expansive dry
snow cover predominates, both $R_c$ and SI maps showed a greater reduction in wet snow coverage compared to S2, highlighting the differing sensitivities of SAR signals to dry snow conditions in these methods compared to S2's multispectral data.

## 4.2   Temporal Dynamics of Snow Melting

In this section, we further applied the proposed method to the collected S1 image time-series between 2017-2021 to generate wet snow maps across all three basins. The generated wet snow maps were resampled from the original SAR image resolution
of $14\times14$m to $30\times30$m before the analysis to speed up the processing. The temporal interval of the time-series was 12 days, the same as the acquisition interval of S1 images. We analyze two key properties of snow cover dynamics derived from the time-series data: the Wet Snow Extent (WSE) and the Snow Melting Duration (SMD). It's worth noting that these properties represent just a subset of the potential insights that can be derived from this dataset.

### 4.2.1   Wet Snow Extent

The temporal patterns and elevation dependencies of WSE across the Hunza, Shigar, and Shyok basins were depicted in Figure 9 (a)~(c). WSE was calculated as the percentage of wet snow pixels within each 100-meter elevation band, offering a granular view of snowmelt progression. Over the five-year period analyzed, a consistent inter-annual pattern was observed in all three basins. Melting typically is initiated by the end of March to early April and is concluded by late September to

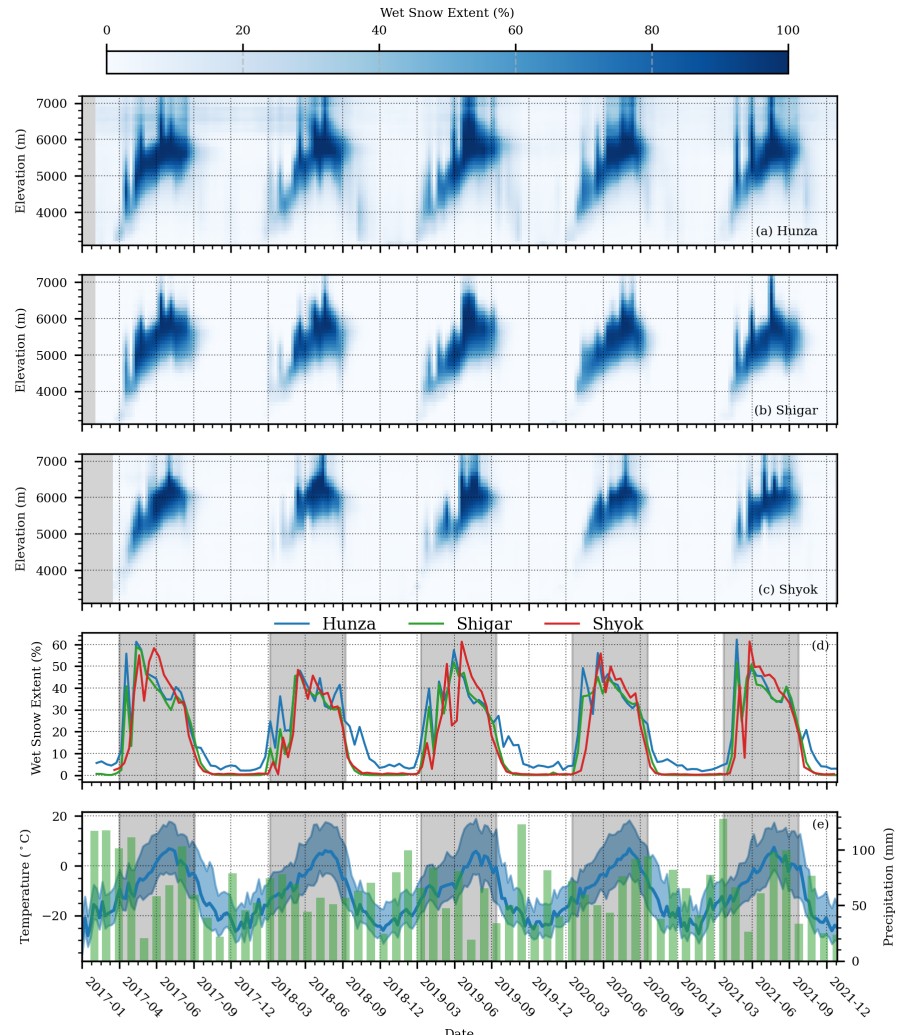

**Figure 9.** Temporal and elevation dependence of WSE in (a) Hunza, (b) Shigar and (c) Shyok, as well as (d) the total WSE of the three basins within the studied period, and (e) temperature and precipitation record obtained from the ERA-5 dataset. In panel (a)∼(c), the WSE is represented by the color scale. In panel (d), blue line indicates the weekly average temperature of the three basins, and the shaded blue indicates the range between the weekly maximum and minimum temperature. Monthly averaged precipitation is shown with the green bar. Shaded gray zone shows the summer of the year. The WSE was calculated as the ratio of wet snow-covered area to the total area within each respective elevation band.

early October. As temperatures rose from spring (i.e., April to May) into summer, the melting front (e.g. the upper and lower elevation boundary of the melting area) ascended along the altitude gradient. Specifically, the lower elevation boundary of wet snow extended upwards as snow at lower altitudes fully melted, while the upper boundary extended to higher altitudes as higher temperatures resulted in melting at greater elevations. In the peak melting months (i.e., July and August), the upper boundary of melting snow reached its maximum altitude before descending, whereas the lower boundary extended to its highest extent and stabilized until the end of the melt season.

Figure 9 (d)∼(e) illustrated the interactions between total WSE, temperature, and precipitation within the region, using data from the ERA-5 reanalysis dataset (Hersbach et al., 2020). The temperature data, averaged weekly, include mean, maximum, and minimum temperatures of air at 2m above the surface of land across the Karakoram region, providing insights into the thermal conditions influencing snow melting. The precipitation data is the accumulated liquid and frozen water falls to the Earth's surface. It was compiled and averaged monthly to complement the temperature analysis by revealing precipitation trends and their impact on snowpack. While the three basins demonstrated similar inter-annual variability, annual discrepancies were pronounced within the time series. For instance, the peak WSE in 2018, around 40%, was notably lower than the approximately 50% observed in other years. This reduction in WSE is related to diminished winter precipitation during the 2017-2018 season, as indicated by the precipitation data. The onset of snow melting aligned well with the period when maximum temperatures rise above freezing, suggesting that peak temperatures were a more sensitive indicator for the onset of snow melting than mean temperatures. These complex inter-annual fluctuations underscore the snowpack's responsiveness to immediate weather conditions, such as temperature spikes and precipitation events.

### 4.2.2 Snow Melting Duration

The SMD reflects the temporal persistence of wet snow cover within a given year, allowing for consistent comparisons across years with varying numbers of observation days. To compute the SMD for each year, we first determined the ratio between days with wet snow cover (M) to the total number of observed days (N) for each pixel. Since the number of observation days (N) varied each year and was typically less than 365, we re-scaled this ratio to a 365-day basis using the formula $(M/N) \times 365$ to standardize the annual average of wet snow cover days and enable consistent comparisons between years.

Figure 10 presents the annual average SMD across the study region from 2017 to 2021. The average SMD displays a pronounced terrain dependency: valley areas at lower altitudes typically exhibit a SMD of fewer than 60 days, while higher altitudes, such as glacier accumulation zones, generally experience SMD exceeding 120 days. In certain high-altitude regions, the wet snow cover can persist for more than 180 days annually.

To assess the temporal and spatial dynamics of SMD, we evaluated the annual fraction of SMD for each basin, as depicted in Figure 11. SMD was categorized into four ranges: 0-60 days (blue), 60-120 days (orange), 120-180 days (green), and 180-240 days (red). In Hunza, the area with a SMD of less than 60 days saw a decline from 2017 to 2019, followed by an increase from 2020 to 2021. This change was inversely related to the 60-120 days category, which expanded from 2017 to 2019 before contracting. The fraction of WSD exceeding 120 days initially decreased from 2017 to 2019 and then increased as the 60-120 days category diminished. A pronounced peak within the 180-240 days range occurred in 2017. The Shigar basin exhibited

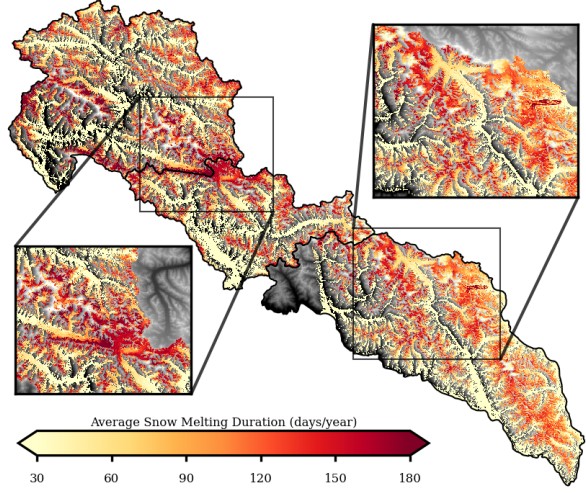

**Figure 10.** Annual average SMD of the study region, calculated based on the five year observation.

more pronounced annual oscillations in SMD. The area with a SMD below 60 days fluctuated around an average of 0.5, while the 60-180 days category mirrored the pattern in Hunza, increasing from 2017 to 2019 before a subsequent decline. The SMD
range of 120-180 days remained relatively stable at about 0.2, with periods exceeding 180 days noted only in 2017 and 2020. The Shyok basin experienced the most substantial temporal variation in SMD. The 0-60 days category showed strong variance around the 0.5 level. The 60-120 days category peaked in 2019, representing a larger fraction ($\sim$0.5) compared to those in Hunza ($\sim$0.3) and Shigar ($\sim$0.35). The 180-240 days range was present exclusively in 2017.

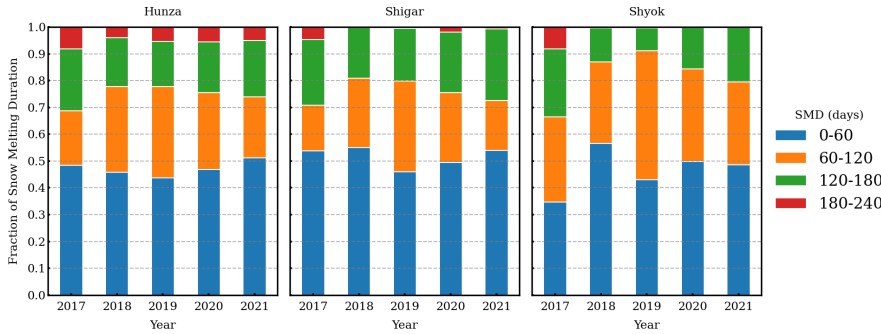

**Figure 11.** Fraction of Snow Melting Duration in the three basins between 2017-2021. SMD was segmented into four categories as represented by different colors.

# 5 Discussion

 ## 5.1 Classification Performance

Compared to the conventional methods where only a single-value threshold are used on the $R_\mathrm{c}$ map, the proposed method has effectively improved the mapping accuracy in the validation. This is primarily attributed to the transformation of $R_\mathrm{c}$ into WSI and the incorporation of TSI.

The GMM enabled a data-driven approach that allowed to adaptively transform the $R_\mathrm{c}$ into WSI based on the local SAR signal responses. As shown in Figure 3, the WSI function for each basin was characterized by distinct center ($x_0$) and slope ($k$) parameters determined from the GMM, indicating that varied local response of SAR backscattering was raised by the diverse nature of wet snow distribution in different basins. This provides the flexibility required for large-scale application, and thus offered an approach for robust wet snow mapping in complex and diverse landscapes.

Furthermore, the proposed method enriched the terrain analysis in snow mapping by incorporating multiple topographical factors (i.e., elevation, slope, and aspect) beyond the traditional use of snow line elevation. The add-on information from slope and aspect enables accurate capture of snow spatial distribution over the complex terrain, which is particularly important in Karakorum's region. As exemplified in Figure 4 for the Hunza basin, the TSI values vary with elevation, aspect, and slope across different times of the year. At the onset of snowmelt, flat and east-facing slopes at altitudes between 3500m a.s.l and 5000m a.s.l exhibited higher TSI values than the other regions, suggesting snow at such area was more susceptible to melting, likely due to solar exposure. As the season progressed into summer, altitude increasingly dictated the snowmelt on flat terrains, while on steeper slopes, aspect also played a significant role. Approaching late autumn, the snow line stabilized at similar altitudes across the two slope classes, yet variations were observed across the aspect with higher TSI values on south-facing slopes (approximate aspect of $75° - 255°$), indicating conditions more conducive to melting. This level of detail in our analysis demonstrated the potential of our method to provide a more comprehensive understanding of wet snow dynamics than analysis using only snow line elevations.

While the proposed method demonstrates strong performance across the three basins in Karakoram, several limitations should be acknowledged. First, the choice of the SI threshold relies on the coefficient determined through sensitivity analysis for the selected study areas. While this approach balances classification performance across the basins, it is not fully dynamic and may require adaptation for larger, more topographically diverse regions, such as global-scale applications. Future work could explore supervised learning models, such as random forests or neural networks, to capture more complex, non-linear relationships between SAR backscatter, topography, and snow conditions, enabling dynamic threshold adaptation.

Second, glacier surfaces present unique challenges for SAR-based snow classification. As shown in results, glacier-specific scattering mechanisms, including contributions from wet debris, bare ice, and supraglacial features, introduce variability in radar backscatter that is not explicitly modeled in the current method. This limitation may lead to underestimation of wet snow on glacier surfaces and highlights the need for further refinement of the method to better handle glacier-specific conditions, potentially by incorporating land surface type information.

Finally, the use of TSI introduces potential bias in regions where topographic conditions deviate significantly from the assumptions underlying its calculation. For instance, low TSI values in ablation zones may lead to underestimation of wet snow, as noted in the comparison with S2 results. While the inclusion of TSI improves overall robustness by integrating terrain characteristics, future studies should evaluate strategies to mitigate these biases, especially in regions with complex topography or unique land surface characteristics.

## 5.2  Implications of Wet Snow Maps

Large scale wet snow maps, especially the ones with high spatial-temporal resolution, have significant implications for hydrological studies, water resource management, and climate impact assessments (Helmert et al., 2018). Snow data obtained from remote sensing and field site stations have proven to be fundamental for the development, calibration, and validation of snowpack, hydrology, and runoff prediction models (Schmugge et al., 2002; Andreadis and Lettenmaier, 2006; Dressler et al., 2006; Griessinger et al., 2019; Cluzet et al., 2024).

Using the wet snow maps generated from the proposed method, our study has extracted and analyzed two critical snow variables, i.e., WSE and SMD, which are crucial for understanding regional snow melting dynamics. The analysis of WSE uncovered detailed patterns of snowmelt changes over time and across elevations, which can provide valuable observations for calibrating snowpack models or forecasting runoff events (Cluzet et al., 2024). The interpretation of SMD highlighted the yearly differences in snow melting duration across the basins, with Hunza exhibiting relative stability and Shyok demonstrating the most variability. A long-term SMD observation record will provide key insights into the changes of the regional climate pattern.

## 6  Conclusions

In this study, we proposed a novel approach for mapping wet snow in complex mountainous regions, such as the Karakoram, by effectively combining the S1 SAR data with topographic information. We first adopted the GMM to adaptively transform the SAR backscattering ratio $R_c$ into the WSI as a robust representation of wet snow under complex surface conditions. Then, we introduced the TSI to capture the likelihood of snow presence influenced by topographic conditions. Validation with S2 snow cover maps demonstrated a notable improvement in the accuracy of wet snow classification.

With the collected time-series of S1 images over the three major water basins in Karakoram, we produced large-scale wet snow maps using the proposed method. The wet snow maps have enabled detailed analysis of crucial snow variables including the WSE and SMD. Analysis on the two variables revealed the dynamic pattern of the temporal-spatial distribution of wet snow in Karakoram, suggesting that the comprehensive dataset produced with this study can offer further enhancement for hydrological model calibrations and validation, thereby ensuring informed water resource management and climate modeling.

Future work involves integrating the approach with in-situ observations and hydrological models to further improve the accuracy and utility of water resource planning tools. Continuing to advance this research would provide results that are greatly beneficial for fostering climate resilience and sustainability in Karakoram.

## Appendix A: Sensitivity Analysis on SI Threshold

We applied a sensitivity analysis on TSI coefficients for optimal SI threshold finding. A series of TSI coefficients were tested using validation images to examine how the TSI coefficient in the SI threshold affected the classification results. Three metrics, including the F1 score, precision and recall, were evaluated and used as the selection criteria. The result (figure A1) showed that the optimal coefficients for Hunza and Shyok were close to 3.5, while a lower coefficient around 2.5 was found to be optimal for Shigar.

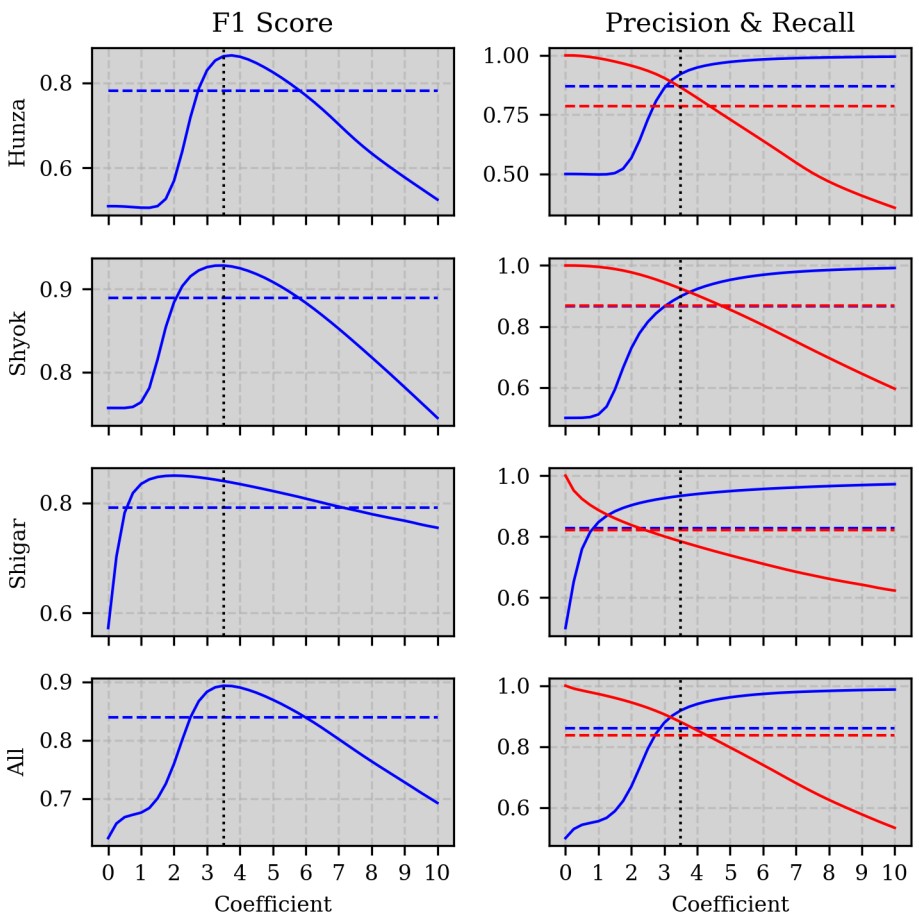

**Figure A1.** Sensitivity analysis on the choice of TSI coefficients for SI thresholds. The left column is for F1 score, and the right column is for Precision (blue) and Recall (red). Dashed lines are results using $R_c = -2\ dB$ as threshold. The dotted vertical line indicates where the coefficient is 3.5.

. S.Li conceived the study idea, devised the methodology, carried out the analysis and wrote the manuscript. All authors contributed to the discussion for methodology development, the interpretation of results and the writing of the manuscript.

. The authors declare no competing interests.

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
