# Peer review of "Mapping Seasonal Snow Melting in Karakoram Using SAR and Topographic Data"

_EGUsphere, 2024_

## Referee Comment (RC1)

**General Comments**

The paper presents an innovative framework for mapping wet snow in the Karakoram region using SAR and topographic data. The integration of the Gaussian Mixture Model (GMM) to determine the Wet Snow Index (WSI) and the calculation of the Topographic Snow Index (TSI) significantly improve the accuracy of snow mapping in complex terrains. And I do appreciate the extensive data sources used by the authors, including Sentinel-1 SAR imagery, Digital Elevation Models (DEM), and Sentinel-2 Level-2A imagery, which ensure robust and detailed mapping of seasonal snow melting. The continuous analysis of Wet Snow Extent and Snow Melting Duration provides valuable insights into the temporal and spatial dynamics of snow melting in the Karakoram region, offering excellent references for water resource management and hydrological modeling, especially in glacial and high mountain areas with complex terrain.

However, I have several concerns regarding the complexity of the methodology and the unclear use of data, as well as the lack of quantified uncertainty analysis. Therefore, I recommend that the paper undergo revisions before it can be considered for publication.

**Here are my detailed comments:**

Line 47, should be "includes"

Line 96, Please specify the spatial resolution of the DEM.

Line 109, Please specify the Sentinel-2 spectral bands used to calculate the LIS in the paper.

Line 122, It is unclear which corrections and preprocessing methods were adopted in the GAMMA software for SAR image processing. Did you apply terrain correction or geometric correction?

Line 131, The focus season of the paper is summer. Why was a multi-year winter value chosen for the reference $\gamma$, Does the reference $\gamma$ have a spatial pattern climatology or a single value for each basin?

Line 146, In the GMM model, which covariance structure did you choose? Were other parameters set by default?

Line 175, I am a little confused about the SI definition. From my understanding, is the WSI first scaled by the TSI, meaning SI = WSI/ TSI? Then you have an SI map, and to further differentiate the wet snow, a threshold of 3.5 is selected. Is this threshold basin-dependent or applied across the entire Karakoram? How do you determine that 3.5 is the optimal threshold, and how do you quantify the uncertainty resulting from this coefficient?

Eq. 7, should be "$\rho_B$"?

Line 184, It would be better if you could clarify the exact thresholds used in the S2 snow cover detection. Currently, the flowchart for S2 is unclear. Specifically: (1) First step to meet the condition NDSI >0.4 and $\rho_B$>0.2; (2) the dark cloud region only defined by $\rho_B$>0.3? Can you explain what is bi-linearly down-sampled red band, from which resolution to which resolution?; (3) and then calculate the snow cover fraction (SCF) at which resolution? (4)

Line 196, is this for dark cloud pixels above the snowline? Based on your description, above snow line, 'no-snow' is defined by NDSI >0.15 and $\rho_B$>0.04 and $\rho_B$<=0.1, and dark cloud is defined by NDSI >0.15 and $\rho_B$>0.1? How consistent are these relaxed thresholds compared to the strict threshold $\rho_B$>0.3 mentioned in 191? Generally, if you are strictly following Gascoin et al. (2019), you can briefly refer to their method, but any changes should be clearly mentioned; otherwise, restate the algorithm clearly and concisely.

Line 203, When you mention summer, which months are included?

Line 204, I am curious about the accuracy in S2 classification, and is it possible to estimate its uncertainty? Generally, in which confidence (and at what elevation) can S2 snow mostly be considered melt snow during summer, given that the accuracy matrix is calculated by comparing snow-free or snow from S2 with no-or-dry snow or melt snow from SI?

Figure 6, How do you explain the remaining mismatch in the ice and melt categories between SI/Rc and S2 detection in the Hunza region?

Figure 7, How do you explain the steeper slope of the snow coverage profile in the SI method compared to the flatter slope in the Rc method?

Line 243, when calculating the SWE and SMD, do you also include the area where a.s.l. over 5500m? Then how to make sure the accuracy above 5500m, given that these areas were excluded in the S2 validation?

Line 256, what level of precipitation and temperature are adopted here, surface or pressure level?

Line 269, does it mean that each pixel you have the wet snow days in total in terms of the whole summer season, then I don't know the purpose of rescale it into 365 days since I guess most of the melting is in summer? It is not appropriate to extend the summer research into the annual. You could just use real wet duration instead.

Line 290, It may not be suitable to use words like "greatly" to describe the improvement if the new classification's improvement is around 5%.

---

## Referee Comment (RC2)

**Review of "Mapping Seasonal Snow Melting in Karakoram Using SAR and Topographic Data"**

Authors: Shiyi Li, Lanqing Huang, Philipp Bernhard, and Irena Hajnsek
MS. No.: egusphere-2024-942
URL: [Mapping Seasonal Snow Melting in Karakoram Using SAR and Topographic Data](Mapping Seasonal Snow Melting in Karakoram Using SAR and Topographic Data)
Referee: Eric Gagliano (egagli@uw.edu)

**General comments**

This paper proposes an improvement to the Nagler family of SAR-based wet snow mapping methods, making use of a gaussian mixture model and newly proposed metrics "wet snow index", "topographic snow index" and "integrated snow index" in order to better separate wet snow from no snow / dry snow by incorporating terrain data and a more dynamic clustering algorithm. This method is evaluated against Sentinel-2 derived snow cover maps, and the authors found improved wet snow identification performance when compared to the Nagler et al., 2016 method. The authors apply their method over the Karakoram, a challenging domain for wet snow identification, to analyze snowmelt dynamics from 2017 to 2021.

I appreciate the authors focus on improving SAR-based wet snow mapping–there is still a lot of room for our community to improve these methods, and this research does a nice job focusing on two important considerations of these algorithms: the incorporation of terrain information and more complex clustering methods. In this paper, I particularly enjoyed the beautiful plots (especially figures 5 and 8!) and the application of this method for time series analysis over multiple melt seasons. This was really great to see, thanks for yall's hard work!

However, I believe the authors should address some important methodological concerns and should also have a chance to improve the overall clarity of the paper. In particular, I think the newly proposed metrics need to be given more motivation and context, especially when it comes to the physical interpretation of different mathematical and methodological operations. I am particularly concerned with the topographic snow index and the implication of its pixelwise multiplication with the wet snow index that yields the integrated snow index. Additionally, some of the analysis and discussion could use some more consideration. Finally, in terms of writing, while I enjoyed the style, I think this paper might benefit from another couple of skims to address issues related to clarity, verb tenses, unnecessary initialisms, misspellings, etc. I've provided more detailed comments and grouped them by subject below. For these reasons, I recommend revisions before publication.

**Specific comments**

**Methods**

Line 128: Do you follow the full Nagler et al., 2016 method, or just the steps listed in your section 3.1? Might want to think about the implications of evaluating against Rc instead of the complete Nagler method.

Line 133: Maybe find some way to motivate the introduction of Rc, or at least give a more thorough explanation of what it is and why we are using it, and why it might be better than VV or VH alone. Perhaps consider adding a line about why a weighting factor W can be helpful in the first place… local incidence angle (LIA) dependence of wet snow detection, etc.

Line 137-139 / 153-154: The wet snow index is a bit unclear to me. Why is Rc insufficient? What is WSI exactly, and what are you claiming it represents? It is not clear to me why you cap the WSI to 10… if WSI were capped at 1, given two Rc values, I see how the absolute differences between their WSI values would be scaled, but don't the relative distances between their two WSI values remain the same? I guess I don't follow how this allows larger Rc variations.

Line 155-157: I think you should motivate why the GMM is necessary…GMM adaptively determines k based on cluster separation, but what does that equate to in practice?

Line 160-172: How do you control for the different wet snow backscatter response at different LIAs / polarizations? Rc combines VV and VH as a function of LIA, which gets at whether VV or VH has more wet snow detection capability at a given LIA. Does using Rc as the basis for WSI muddy the water in some way because each pixel is measuring a different proportion of VV and VH data?

Line 160 / 174-176: I am concerned with TSI and its implications on SI / your final wet snow classification. I understand your scaling intent, but I think you may be biasing your wet snow detection across different topographic bins. If my understanding is correct, you calculate WSI at each pixel for an entire scene, then group pixels into the topographic bins, and then take the median WSI in each bin which gives you TSI. Then you get pixelwise SI values by multiplying WSI by TSI. Does this mean you are implicitly assuming something about the relative proportions of wet snow and dry / no snow in each bin? Consider two scenarios: for a lower elevation bin, assume 10% of pixels have wet snow in the summertime SAR image. The majority of the WSIs should be close to 0, and your TSI will likely be low. For those pixels in the bin that are indeed wet, their WSI value now gets multiplied by a very low TSI, and now has a much greater chance of going undetected as wet due to a low SI. Now instead consider the case for a much higher elevation bin, in which the summertime SAR image shows 80% of pixels containing wet snow. The TSI will likely be closer to 10, and so even a pixel with a WSI of 1.5 will end up with an SI of 15. Given how sharp the TSI changes in Figure 4, this could be concerning because it seems like TSI forces pixels towards the classification of the majority of

pixels in their topographic bin, potentially overriding each pixel's individual backscatter response. I believe this can be seen in Figure 7… notice how below 5000m, you almost always are underpredicting snow cover relative to S2 snow cover. This consistent low bias / false negatives at lower elevations would be expected with the issue mentioned above. *I would really like to see a plot of F1 score or other classification metric aggregated by your TSI bins (same axes as figure 4)--this could reveal the influence of TSI on your classification performance*. To address this issue with minimal changes to your methodology (if you are truly confident in your method despite these concerns), you might consider some sort of figure that shows spatial maps of Rc, WSI, TSI, SI, SI classification, and S2 snow cover map all included, ideally all with histograms? So something similar to figure 5 but adding in WSI and TSI maps and histograms. I think this may help readers build intuition for WSI and TSI, and especially how combining these via multiplication into SI ultimately improves your classification.

Line 179-181: I'm confused on how 3.5 was chosen, it feels relatively arbitrary. Is there an analysis of different coefficients and how they affect the results? Does the coefficient vary across basins, and does it vary annually / based on summer image? If a claim of this paper is to be able to make snow detection algorithms more robust to different locations, will this cutoff have to be chosen dynamically?

Line 268-270: Hmmm, can you just scale to 365 like this? Is this a common practice? I would be concerned about sampling, I'm sure there might be a better way to quantify SMD, maybe day of (last day of wet snow - day of first wet snow)?

**Interpretation of results**

Table 2: The adaptive SI thresholds seem to be quite different for these three study sites that are relatively close to each other. It would be helpful to see more interpretation on this.

Line 203-206: This gets at two important limitations… it is possible at the date you select, not all snow pixels are wet, and even across scenes where snow is wet at all pixels, your choice of summer date likely influences how much liquid water is in the snowpack and therefore backscatter response (Baghdadi et al., 2000; Karbou et al., 2021). My guess is that the backscatter drop is more pronounced during certain portions of the melt season. Given this, I think it is important to see how your adaptive SI thresholds change with your choice of summer scene.

Line 211-213: It might be helpful to mention that the noisy patterns over the glaciers are harder to ascribe to the variables you list because of glacial movement. Not sure of the glacier speeds in this area, but it is possible that you are seeing large backscatter changes due to glacier movement that occurred between your winter scenes and summer scene.

Line 213-215: I'm not sure it's correct to claim SI reduces the noisy pattern because of the rescale to WSI, I think it's just the multiplication of TSI which is essentially smoothing the pixels in those particular bins, right?

Line 216: You could consider showing these distributions you mention, it would be really helpful to contextualize SI ranges.

Line 217-219: I'm not too sure you can say this is effective over glacier surfaces… you predict very little wet snow on glaciers, but wouldn't we expect the glacier surface to have a significant amount of wet snow or wet ice in the summer, which should significantly drop the backscatter and be classified as wet snow (even though technically it could be wet ice)? For SAR identification of glacier melt, I would check out Scher et al., 2021. I am concerned about this, especially because in 5c not a lot of glacier area in the insets seem to be identified with wet snow. I think the TSI plays a large role in reducing the SI in these locations (especially with the two slope bins). It may be worthwhile to consider removing glaciers from your analysis, due to these reasons and the fact that significant glacier motion has likely occurred between the winter and summer scenes, so the area in a pixel in your reference image is not the same area in the same pixel in your summer image.

Figure 8: Really pretty plot! For each elevation band, I'm guessing WSE is (wet snow pixels / total number of pixels), not (wet snow pixels / total number of snow pixels), right?

Line 245-288: It would be great to know how sensitive the WSE and SMD analysis is to the choice of your baseline winter composite and summer image. Do you apply this algorithm to one image and use those SI thresholds? Or do you do it separately for every image in the time series? If it is every image separately, it would be great to know how your WSI/TSI/SI distributions change, and how your SI threshold changes.

**Clarity**

Figure 1: Are the black boxes scene footprints? Maybe make this explicit, and if you keep the boxes, maybe label them or color code them with relative orbit information and add a legend? And I think your colorbar may be flipped?

Line 81: I'm not too sure what this sentence is getting at, perhaps reword?

Figure 4: Not sure if a red and blue diverging color map is most appropriate here. I feel like the white values don't play an especially important role (a value of ~5 for TSI doesn't seem particularly important), and the color map could cause confusion with the reader thinking it is the same quantity as Figure 5's Rc plot (though I think the red and blue color map for that Rc plot is great!).

Line 179: Clarify "...where 3.5 implied the condition applied to the TSI…"

Line 225: What percentage of your pixels were excluded, and why was 5500 m a.s.l. chosen?

Line 237: I would clarify what you mean by "...comparative sensitivity of the SAR signal to dry snow in both methods."

Figure 5: It is not clear where the S2 snow cover map is coming from. Is this output from the let-it-snow algorithm? If so, where is this ice/water class coming from? Also, it might be helpful to say in the caption that these are all three basins.

Line 289-291: Definitely a really neat approach, but maybe temper your claim of great improvement here, or provide the specific stats.

Line 301-304: I think this is a more fair interpretation of TSI! It should be helpful for understanding aggregate backscatter response of topographic bins, I'm not sure about the added use of using it as a scale factor for WSI.

Line 316-318: Provide citation

Line 324: You haven't previously talked about mitigating uncertainty, so I would avoid introducing this idea in the conclusion.

Line 324-325: I feel like the purpose/definition of TSI shifts around a bit in this paper. Is it supposed to take into account the influence of topography on snow presence or the influence of topography on the backscatter response of wet snow?

**Technical corrections**

Line 5: For the abstract, I would use "a" instead of "the" in front of both Wet Snow Index and Topographic Snow Index so it's clear to that these are metrics that you are introducing
Line 10-11: Remove "the" in between under and climate
Figure 1: "tree" -> "three"
Line 47: "inlucdes", also check tenses / plural of this line
Line 53: Remove "of"
Line 67: "meting" -> "melting"
Line 67: "watersheds" -> "watershed"
Line 69-72: Make verb tenses agree
Line 89: "snow melting changes" -> "snowmelt"
Line 108: "the" -> "a"
Line 116-118: unclear
Line 122: "backscattering intensity images" -> "backscatter images"
Line 123: "firstly" -> "first"
Line 131: add "is" before "the reference image"
Line 216-217: Check tenses
Line 218: "well" -> "good"
Line 221: Consider rewording "greatly false positives"
Table 3: "corresponded" -> "correspond"

Line 278: Consider "This trend" -> "This change"
Line 280: Maybe avoid "significant", might be confused for an actual test of significance
Line 284: Remove "significant"
Line 305: "was" -> "were"
Line 306: Should there be a dash between the two aspects here?
Line 311: "Management" -> "management"
Line 312: Remove "been"
Line 319: Remove "significant"
Line 320: "observation" -> "observation record"
Line 325-326: Wording of this sentence is a bit confusing, consider rewriting and remove "significant" unless some sort of hypothesis testing was done.
Line 328-329: Remove "such as" since these were the only two variables analyzed in the paper.
Many lines: Check usage of initialisms (GMM, BOA, RMSE, MSE, WSI, SVM, RF, ML, TSI, etc). Often initialisms are introduced multiple times, and some initialisms are introduced and never used (or sometimes used once)
Many lines: Tenses are often inconsistently used, I would skim over to make sure they agree!

**References**

Baghdadi, N., Gauthier, Y., Bernier, M., & Fortin, J.-P. (2000). Potential and limitations of RADARSAT SAR data for wet snow monitoring. *IEEE Transactions on Geoscience and Remote Sensing*, *38*(1), 316–320. IEEE Transactions on Geoscience and Remote Sensing. https://doi.org/10.1109/36.823925

Karbou, F., Veyssière, G., Coleou, C., Dufour, A., Gouttevin, I., Durand, P., Gascoin, S., & Grizonnet, M. (2021). Monitoring Wet Snow Over an Alpine Region Using Sentinel-1 Observations. *Remote Sensing*, *13*(3), Article 3. https://doi.org/10.3390/rs13030381

Nagler, T., Rott, H., Ripper, E., Bippus, G., & Hetzenecker, M. (2016). Advancements for Snowmelt Monitoring by Means of Sentinel-1 SAR. *Remote Sensing*, *8*(4), Article 4. https://doi.org/10.3390/rs8040348

Scher, C., Steiner, N. C., & McDonald, K. C. (2021). Mapping seasonal glacier melt across the Hindu Kush Himalaya with time series synthetic aperture radar (SAR). *The Cryosphere*, *15*(9), 4465–4482. https://doi.org/10.5194/tc-15-4465-2021

---

## Author Comment (AC1)

Response to RC2

**Mapping Seasonal Snow Melting in Karakoram Using SAR and Topographic Data**

Discussion: https://doi.org/10.5194/egusphere-2024-942

Comments from the reviewers are given in black.

Our responses are given in blue.

General Comments

This paper proposes an improvement to the Nagler family of SAR-based wet snow mapping methods, making use of a gaussian mixture model and newly proposed metrics "wet snow index", "topographic snow index" and "integrated snow index" in order to better separate wet snow from no snow / dry snow by incorporating terrain data and a more dynamic clustering algorithm. This method is evaluated against Sentinel-2 derived snow cover maps, and the authors found improved wet snow identification performance when compared to the Nagler et al., 2016 method. The authors apply their method over the Karakoram, a challenging domain for wet snow identification, to analyze snowmelt dynamics from 2017 to 2021.

I appreciate the authors focus on improving SAR-based wet snow mapping–there is still a lot of room for our community to improve these methods, and this research does a nice job focusing on two important considerations of these algorithms: the incorporation of terrain information and more complex clustering methods. In this paper, I particularly enjoyed the beautiful plots (especially figures 5 and 8!) and the application of this method for time series analysis over multiple melt seasons. This was really great to see, thanks for yall's hard work!

However, I believe the authors should address some important methodological concerns and should also have a chance to improve the overall clarity of the paper. In particular, I think the newly proposed metrics need to be given more motivation and context, especially when it comes to the physical interpretation of different mathematical and methodological operations. I am particularly concerned with the topographic snow index and the implication of its pixelwise multiplication with the wet snow index that yields the integrated snow index. Additionally, some of the analysis and discussion could use some more consideration. Finally, in terms of writing, while I enjoyed the style, I think this paper might benefit from another couple of skims to address issues related to clarity, verb tenses, unnecessary initialisms, misspellings, etc. I've provided more detailed comments and grouped them by subject below. For these reasons, I recommend revisions before publication.

**Response:**

Thank you for the constructive and thoughtful feedback. We greatly appreciate your recognition of our contributions to SAR-based wet snow mapping, as well as your positive comments on our methods and visual presentation. We are committed to addressing the methodological concerns you raised and clarifying the motivations and interpretations of our proposed metrics. Your insightful discussion of the potential limitations of our method has been especially valuable in improving the manuscript. In response, we have added more detailed explanations of our methodology and clearer interpretations of our proposed metrics. Additionally, we have extended the discussion section to explicitly address the limitations of our approach, including the challenges associated with glacier surfaces, the influence of TSI on classification, and the need for dynamic thresholding in larger-scale applications. These revisions aim to enhance the clarity, coherence, and overall quality of the paper. Thank you again for your valuable feedback and for helping us refine our work.

**Specific comments:**

**Methods**

Line 128: Do you follow the full Nagler et al., 2016 method, or just the steps listed in your section 3.1? Might want to think about the implications of evaluating against Rc instead of the complete Nagler method

The method proposed by Nagler et al., 2016 primarily involves three key steps: (1) preprocessing VV and VH backscatter images for data preparation and creating a reference image by averaging multiple snow-free images, (2) calculating the backscatter ratio between each image and the reference image for the VV and VH channels and combining these with LIA-dependent weighting factors to derive the composite ratio (Rc), and (3) applying a threshold of -2 dB to Rc for distinguishing wet snow from dry snow.

In our approach, we adhered to these very same steps. As this paper does not aim to reintroduce the full details of Nagler's method, which is well established within the community, we did not elaborate extensively on the Rc generation process. However, Section 3.1 includes the necessary processing steps and parameters to illustrate how we generated the backscatter ratio. We also emphasized our specific selection of reference images in the challenging Karakoram region in the revised text.

For the comparison experiments, we evaluated our method against the threshold-based approach of Rc < -2 dB, ensuring a direct comparison with the Nagler method.

The revised text emphasizing the choice of reference images are as follows: "*Note that this differs from other alpine regions, such as the Alps, where summer months are often used as the reference due to snow-free conditions. In the Karakoram, using winter images as a reference effectively highlights the contrast in backscattering intensity between summer wet snow (lower intensity due to water absorption) and winter dry snow.*"

Line 133: Maybe find some way to motivate the introduction of Rc, or at least give a more thorough explanation of what it is and why we are using it, and why it might be better than VV or VH alone. Perhaps consider adding a line about why a weighting factor W can be helpful in the first place... local incidence angle (LIA) dependence of wet snow detection, etc.

Thank you for the suggestion. The concepts of the composite ratio Rc and the use of weighting factors (W) based on local incidence angle (LIA) are well-established and detailed in the Nagler et al., 2016 paper. In the revision, to enhance the clarity of our paper, we added a brief explanation and motivation for the use of Rc and the benefit of LIA-dependent weighting at the beginning of the section as the following:

*"With the preprocessed $\gamma^0$ images, we derived the composite backscatter ratio Rc following the method proposed by Nagler et al., 2016 . The Rc metric combines the backscatter ratios from both the VV and VH channels to comprehensively assess surface condition changes associated with snow melting. This approach incorporates a weighting factor W, which is determined by the local incidence angle (LIA, θ), to account for variations in backscatter due to differing incidence angles. This adjustment enables more robust snow melt detection across varying terrain geometries".*

Line 137-139 / 153-154: The wet snow index is a bit unclear to me. Why is Rc insufficient? What is WSI exactly, and what are you claiming it represents? It is not clear to me why you cap the WSI to 10... if WSI were capped at 1, given two Rc values, I see how the absolute differences between their WSI values would be scaled, but don't the relative distances between their two WSI values remain the same? I guess I don't follow how this allows larger Rc variations.

Thank you for your comments and insightful question. While Rc alone effectively reflects surface condition changes, it can be sensitive to local variations and lacks adaptive boundaries for wet snow classification. Rondeau-Genesse et al. (2016) suggested using a logistic function to convert Rc into a probabilistic space, allowing a soft threshold for wet snow classification. However, this approach requires empirical parameter selection for the logistic function and threshold values. In our work, we utilized GMM to dynamically determine the logistic function parameters, transforming Rc into the Wet Snow Index (WSI) to represent the probability of wet snow presence.

A logistic function typically outputs values between 0 and 1 with the carrying capacity L=1 (indicating probability). Here we chose to rescale the WSI range to 0–10 by setting L = 10 to allow a better combination with the Topographic Snow Index (TSI) used in later steps. This rescale can amplify the distinction between two Rc values. For example, if two points originally had WSI values of 0.1 and 0.9, the rescaled WSI values would be 1 and 9, thereby showing larger discrepancies.

We added the following sentences in the revised manuscript to address the motivation of replacing Rc with the WSI to line 137-139: "*While Rc alone effectively indicates surface condition changes, it can be sensitive to local variations and does not inherently incorporate adaptive boundaries for wet snow classification. Therefore, instead of directly applying a threshold to Rc for wet snow classification, we propose using a GMM to convert Rcinto a Wet Snow Index (WSI) to have a probabilistic measure that better captures the varying conditions of wet snow across different terrains. By leveraging the density distribution of Rc values learned through the GMM, the WSI enables a dynamic scaling of the classification based on the underlying distribution of Rc values.*"

To address the second concern regarding the re-scaling of the WSI values, we re-wrote equation (5) as:

$$WSI = \frac{L}{1 + e^{k(x - x_0)}}$$

and we rewrote line 154 as: "*where k the slope factor, $x_0$ the logistic curve's midpoint and L the carrying capacity representing the supremum of the function. Here the carry capacity L was set to 10 to amplify the differences between pixels of different Rc values.*"

Line 155-157: I think you should motivate why the GMM is necessary…GMM adaptively determines k based on cluster separation, but what does that equate to in practice?

The GMM was introduced as an unsupervised model to adaptively determine the optimal parameters for the WSI logistic function. It allows the algorithm to choose the proper parameters for the conversion between Rc and WSI, which better accommodates the local variation of signal caused by challenging surface conditions. The slope factor "k" controls the slope of the curve, thereby controlling the sensitivity of the WSI to the difference between wet and dry (or no) snow.

In the revision, we further clarified the motivation and the purpose of parameter K with the revised sentences in line 155-160 as the following: "*In the WSI logistic function, the slope factor k is determined by the separation between the two clusters, providing a flexible and adaptive control over the sensitivity of WSI to the difference between snow conditions. In the case when the two clusters are perfectly distinct from each other (i.e. $|\mu_1 - \mu_2| \gg (\sigma_1 + \sigma_2)$), the WSI would be transformed into a step function and thus effectively act as a single-value hard-threshold at $x_0$. Therefore, the Nagler's method can be taken as a special case under this assumption with $x_0 = -2dB$. Contrarily, the mixed clusters (i.e. $|\mu_1 - \mu_2| \ll (\sigma_1 + \sigma_2)$) would lead to progressively flattened WSI and soft segmentation boundaries. A similar form of logistic function was proposed by \citet{rondeau-genesse_monitoring_2016}, which was determined with empirical parameters and was used as soft thresholds to replace the hard-threshold of -2 dB on Rc. In our approach, the GMM allows for an adaptive choice of the parameter k based on the distribution density of Rc, thereby enabling flexible and robust applications in large scale.*"

Line 160-172: How do you control for the different wet snow backscatter response at different LIAs / polarizations? Rc combines VV and VH as a function of LIA, which gets at whether VV or VH has more wet snow detection capability at a given LIA. Does using Rc as the basis for WSI muddy the water in some way because each pixel is measuring a different proportion of VV and VH data?

Thanks for this question. Using Rc as the basis for WSI does not introduce confusion in terms of proportionality between VV and VH. The LIA-dependent weighting is applied to combine VV and VH because the two polarizations have different sensitivities to snow wetness at varying LIAs. This weighting allows Rc to effectively capture backscatter changes due to snow conditions while minimizing the influence of incidence angle. Rather than reflecting a "proportion" of VV or VH, Rc leverages the most effective part of each signal based on LIA, resulting in a more robust, LIA-independent representation of backscatter change.

Line 160 / 174-176: I am concerned with TSI and its implications on SI / your final wet snow classification. I understand your scaling intent, but I think you may be biasing your wet snow detection across different topographic bins. If my understanding is correct, you calculate WSI at each pixel for an entire scene, then group pixels into the topographic bins, and then take the median WSI in each bin which gives you TSI. Then you get pixelwise SI values by multiplying WSI by TSI. Does this mean you are implicitly assuming something about the relative proportions of wet snow and dry / no snow in each bin? Consider two scenarios: for a lower elevation bin, assume 10% of pixels have wet snow in the summertime SAR image. The majority of the WSIs should be close to 0, and your TSI will likely be low. For those pixels in the bin that are indeed wet, their WSI value now gets multiplied by a very low TSI, and now has a much greater chance of going undetected as wet due to a low SI. Now instead consider the case for a much higher elevation bin, in which the summertime SAR image shows 80% of pixels containing wet snow. The TSI will likely be closer to 10, and so even a pixel with a WSI of 1.5 will end up with an SI of 15. Given how sharp the TSI changes in Figure 4, this could be concerning because it seems like TSI forces pixels towards the classification of the majority of pixels in their topographic bin, potentially overriding each pixel's individual backscatter response. I believe this can be seen in Figure 7... notice how below 5000m, you almost always are underpredicting snow cover relative to S2 snow cover. This consistent low bias / false negatives at lower elevations would be expected with the issue mentioned above. *I would really like to see a plot of F1 score or other classification metric aggregated by your TSI bins (same axes as figure 4)--this could reveal the influence of TSI on your classification performance*. To address this issue with minimal changes to your methodology (if you are truly confident in your method despite these concerns), you might consider some sort of figure that shows spatial maps of Rc, WSI, TSI, SI, SI classification, and S2 snow cover map all included, ideally all with histograms? So something similar to figure 5 but adding in WSI and TSI maps

and histograms. I think this may help readers build intuition for WSI and TSI, and especially how combining these via multiplication into SI ultimately improves your classification.

Thank you for your in-depth analysis of the TSI and its role in our wet snow classification. You raise a valid point regarding the potential bias introduced by multiplying TSI with WSI, particularly the possibility of false negatives in low TSI bins and false positives in high TSI bins.

We acknowledge that, without a supervised approach, some degree of bias is inevitable when applying TSI to different topographic regions. To address this concern, we have taken specific steps to mitigate these biases in the method design:

- **Use of Median for TSI Calculation**: Rather than using the mean WSI in each topographic bin, we calculate the TSI as the median WSI within each bin. This choice reduces the impact of extreme WSI values within each bin, making TSI more robust to variations and less likely to skew toward outliers, thus minimizing the bias you mentioned.

- **Adjusted Threshold for Final Classification**: We selected a threshold of 3.5 × WSI, rather than the midpoint value of 5 from the TSI range, for our final classification. This adjustment helps control the bias by moderating TSI's influence on WSI in both low and high TSI regions, balancing the potential for overestimation and underestimation in different elevation ranges. More about the choice of 3.5 can be found in the next comment.

We have plotted classification metrics across topographic bins (F1 score, precision, and recall) as you suggested in Figure 1~3 . The results show that, while there is a lower recall in lower elevation regions (indicating some false negatives) and lower precision in higher elevation regions (indicating some false positives), the overall F1 score remains consistent across elevation bands. Additionally, the results demonstrate increased precision in the low-middle elevation range and improved recall at high elevations, which suggests that the multiplication of TSI with WSI achieves a balanced trade-off between overestimation and underestimation.

We have added the following paragraph in Section 5.1 to discuss the potential bias introduced by the TSI:

"*Finally, the use of TSI introduces potential bias in regions where topographic conditions deviate significantly from the assumptions underlying its calculation. For instance, low TSI values in ablation zones may lead to underestimation of wet snow, as noted in comparison with S2 results. While the inclusion of TSI improves overall robustness by integrating terrain characteristics, future studies should evaluate strategies to mitigate these biases, especially in regions with complex topography or unique land surface characteristics.*"

We will add relevant figures in the supplement. Thank you again for your insightful feedback, which has allowed us to clarify and further validate our methodological choices.

[Figure]

*Figure 1. F1 score aggregated by topographic bins in Hunza. Top rows are with slope < 20 degrees, and bottom rows are with slope > 20 degrees. Missing values are indicated with gray.*

[Figure]

Figure 2. Recall metric aggregated by topographic bins in Hunza. Same arrangement as Figure 1.

[Figure]

*Figure 3. Precision metric aggregated by topographic bins in Hunza. Same arrangement as Figure 1.*

Line 179-181: I'm confused on how 3.5 was chosen, it feels relatively arbitrary. Is there an analysis of different coefficients and how they affect the results? Does the coefficient vary across basins, and does it vary annually / based on summer image? If a claim of this paper is to be able to make snow detection algorithms more robust to different locations, will this cutoff have to be chosen dynamically?

Thank you for pointing out the confusion regarding the choice of the coefficient for SI thresholds. To demonstrate the considerations behind the coefficient choice, we will add the sensitivity analysis in the manuscript.

As shown in Figure 4, we evaluated the sensitivity of different classification metrics (F1 score, precision and recall) on the choice of the coefficient. This analysis showed that a value of 3.5 yields optimal metrics across the three basins. Notably, while Hunza and Shyok exhibit similar responses to this coefficient, Shigar reaches its optimum around 2.5. However, since this analysis was based on selected validation dates, we chose 3.5 as an overall coefficient to avoid overfitting.

Regarding seasonal variation, the coefficient does not change annually or based on individual summer images. This is because WSI is derived using the GMM trained on randomly selected samples from all summer images. Thus, WSI represents an aggregated measure for each basin across observed years, making TSI an overall topographic representation rather than a season-specific one.

As for dynamic cutoffs, the need depends on the application scale. In our study, the three basins are geographically close but exhibit unique local topographic features. We performed sensitivity analysis to evaluate how the threshold multiplier affects accuracy. The results in Figure 4 show that a basin-specific optimal coefficient could be used if overfitting were not a concern. For larger, more topographically diverse areas, we anticipate that a more dynamic cutoff would be necessary. One approach could be to apply supervised learning models, such as random forests or neural networks, to capture the non-linear relationships between SAR and topographic features. While this is beyond the scope of the current study, we intend to explore these methods in future research.

To better explain the choice of the coefficient, and discuss the limitation of our method, we revised the section 3.4 in the manuscript as the following:

```

*As illustrated in Figure 2, the final step generated an integrated Snow Index (SI) map by performing pixel-wise multiplication of WSI and TSI. This multiplication scales the WSI by incorporating terrain characteristics, thereby linking the observed SAR backscattering ratio directly with terrain properties.*

*In order to classify the integrated SI into binary snow maps, it is crucial to apply an adaptive threshold that accounts for the variation in topographic features across different basins. The variation in SAR backscatter response within a basin is inherently handled by the GMM when deriving the WSI. In contrast, the TSI is time-varying and basin-specific, requiring an optimal coefficient to condition the SI for classification. To determine this coefficient, we performed a sensitivity analysis, evaluating F1-score, precision, and recall across different values using the S2 validation snow map.*

*The results, shown in Figure 5, demonstrate that Hunza and Shyok exhibit similar responses, with optimal coefficients close to 3.5, while Shigar reaches its optimum at approximately 2.5. However, to avoid overfitting to specific basins or validation dates, we*

*selected 3.5 as an overall coefficient to balance classification performance across all basins. This coefficient also reflects a moderate threshold applied to the TSI to determine the overall SI threshold for each basin.*

*The threshold is calculated using the following equation:*

$$SI\ threshold = 3.5 \times WSI|_{R_c=-2}$$

*where $WSI|_{R_c=-2}$ represented the WSI at a backscatter ratio Rc = -2dB for each basin. This value is basin-specific, allowing the threshold to adapt based on each basin's distinct characteristics. Together, these conditions form an integrated, basin-adaptive thresholding mechanism, combining SAR backscatter and topographic information into a single index to determine the SI threshold.*

*It is important to note that while the SI threshold is basin-specific, it is time-independent. The WSI is derived from a GMM trained on samples collected from multiple summer scenes over several years, ensuring that it represents an aggregated measure for each basin and is not tied to individual scenes or seasons. This design ensures robustness to seasonal variations in liquid water content and enables consistent application across different validation dates.*

` ` `

We also added an discussion about the limitation of the dynamic cut-off in Section 5.1 as follows:

` ` `

*While the proposed method demonstrates strong performance across the three basins in Karakoram, several limitations should be acknowledged. First, the choice of the SI threshold relies on the coefficient determined through sensitivity analysis for the selected study areas. While this approach balances classification performance across the basins, it is not fully dynamic and may require adaptation for larger, more topographically diverse regions, such as global-scale applications. Future work could explore supervised learning models, such as random forests or neural networks, to capture more complex, non-linear relationships between SAR backscatter, topography, and snow conditions, enabling dynamic threshold adaptation.*

` ` `

[Figure]

*Figure 4. Sensitivity analysis on the choice of coefficients for SI thresholds. The left column is for F1 score, and the right column is for Precision (blue) and Recall (red). Dashed lines are results using Rc=2dB as threshold. The dotted vertical line is 3.5.*

Line 268-270: Hmmm, can you just scale to 365 like this? Is this a common practice? I would be concerned about sampling, I'm sure there might be a better way to quantify SMD, maybe day of (last day of wet snow - day of first wet snow)?

Thank you for raising your concern about this point. In the collected dataset, we had a varying number of observed days for each year, resulting in a time-series with uneven intervals. If we use the last day minus the first day, we can not guarantee that the first and last day in the collected data are indeed the exact first and last day. When we scale the observations on a 365-day annual basis, the observation period are standardized to an annual basis and therefore allows for consistent comparison between different years.

We revised the paragraph as follows to elaborate on this point:

"*The SMD reflects the temporal persistence of wet snow cover within a given year, enabling consistent comparisons across years with varying numbers of observation days. To compute the SMD for each year, we first determined the ratio of days with wet snow cover (M) to the total number of observed days (N) for each pixel. Since the number of observed days (N) varied each year and was typically less than 365, we rescaled this ratio to a 365-day basis using the formula (M/N) ×365. This standardization allows us to calculate the annual average of wet snow cover days, facilitating consistent comparisons between years.*"

**Interpretation of results**

Table 2: The adaptive SI thresholds seem to be quite different for these three study sites that are relatively close to each other. It would be helpful to see more interpretation on this.

As noted in our response to the comment on Line 179-181, we will include the sensitivity analysis as shown in Figure 4 to clarify the choice of SI thresholds across the basins.

Line 203-206: This gets at two important limitations... it is possible at the date you select, not all snow pixels are wet, and even across scenes where snow is wet at all pixels, your choice of summer date likely influences how much liquid water is in the snowpack and therefore backscatter response (Baghdadi et al., 2000; Karbou et al., 2021). My guess is that the backscatter drop is more pronounced during certain portions of the melt season. Given this, I think it is important to see how your adaptive SI thresholds change with your choice of summer scene.

Thank you for your thoughtful comment. We agree that variations in liquid water content in the snowpack can impact backscatter response. To make the WSI robust against these changes, we fitted the GMM for each basin using samples taken from multiple summer scenes across several years, thereby determining stable parameters for the WSI. This ensures that the WSI is specific to each basin but independent of the particular summer dates chosen for validation.

Regarding the TSI, while it varies over time, we set a fixed value of 3.5 in the SI threshold calculation, resulting in an SI threshold of $3.5 \times WSI_{Rc=2}$ . This threshold is basin-adaptive but time-independent, enabling seasonal robustness within each basin. Consequently, our basin-adaptive SI threshold remains consistent regardless of the chosen validation scenes, and the influence of liquid water variability on backscatter response is inherently addressed through the multi-year GMM fitting.

In the revision, we added the following sentences for clarification: "*In the three basins, adaptive SI thresholds were used to generate the snow classification maps. The threshold*

*values for each basin are also reported in Table 2. These thresholds provided basin-adaptive and time-independent classification boundaries to distinguish wet snow and dry snow or snow-free pixels.*"

Line 211-213: It might be helpful to mention that the noisy patterns over the glaciers are harder to ascribe to the variables you list because of glacial movement. Not sure of the glacier speeds in this area, but it is possible that you are seeing large backscatter changes due to glacier movement that occurred between your winter scenes and summer scene.

Thank you for this insightful suggestion. We agree that glacier movement could contribute to the observed noisy patterns in the Rc map over glacier surfaces. In addition to the mixing of wet snow, water, melting ice, and vegetation within a pixel, glacial movement between the winter and summer scenes may cause additional backscatter variability, further increasing uncertainty in these regions.

In the revision (Section 4.1), we added more interpretation regarding results over glacier surfaces. Please find detailed discussion and the revision in the reply to the comment on Line 217-219.

Line 213-215: I'm not sure it's correct to claim SI reduces the noisy pattern because of the rescale to WSI, I think it's just the multiplication of TSI which is essentially smoothing the pixels in those particular bins, right?

Thank you for this observation. We have revised the sentence to clarify this point as you suggested:

"*On the Integrated SI map, the noisy pattern was greatly reduced. This improvement is primarily attributed to the incorporation of topographic information through TSI, which smooths SI values within each topographic bin.*"

Line 216: You could consider showing these distributions you mention, it would be really helpful to contextualize SI ranges.

Thanks for the suggestion. We added the following histogram of Rc, WSI, TSI and SI in the three basins in the revised manuscript.

[Figure]

*Figure 5.Distribution of Rc, WSI, TSI, and SI of the three basins (left: Hunza, middle: Shyok, right: Shigar). The x-axis of Rc is inverted to align with the indexes (negative Rc corresponds to higher probability of wet snow).*

Line 217-219: I'm not too sure you can say this is effective over glacier surfaces... you predict very little wet snow on glaciers, but wouldn't we expect the glacier surface to have a significant amount of wet snow or wet ice in the summer, which should significantly drop the backscatter and be classified as wet snow (even though technically it could be wet ice)? For SAR identification of glacier melt, I would check out Scher et al., 2021. I am concerned about this, especially because in 5c not a lot of glacier area in the insets seem to be identified with wet snow. I think the TSI plays a large role in reducing the SI in these locations (especially with the two slope bins). It may be worthwhile to consider removing glaciers from your analysis, due to these reasons and the fact that significant glacier motion has likely occurred between the winter and summer scenes, so the area in a pixel in your reference image is not the same area in the same pixel in your summer image.

Thank you for your detailed feedback and for highlighting Scher et al., 2021, which provides valuable insights into the complexities of glacier surface scattering. We agree that glacier surfaces represent a challenging scenario for wet snow detection and that further investigation is needed to evaluate how the SI responds to glacier-specific conditions.

Based on Scher et al., 2021, the scattering response in the ablation zone is highly complex and not fully characterized. Processes such as radar backscatter brightening during refreeze and the presence of supraglacial features (e.g., crevasses, suncups, debris cover) contribute to highly variable scattering mechanisms over short distances. These complexities, combined with snow-off conditions, likely result in significant backscatter variability that is challenging to model.

In our comparison between S1 and S2 results, we treated wet snow and wet ice/water as distinct classes. The S2 map incorporates a threshold on the NIR band to separate ice/water from snow. For example, in the insets of Figure 5(d), large areas of glacier ablation zones are classified as ice/water. These regions correspond to snow-free classifications in Figure 5(c). Based on the S2 result, we interpret these areas as being under snow-off conditions, where SI correctly identifies them as snow-free, even though they may contain wet ice or melting glacier surfaces.

We acknowledge that the TSI plays a significant role in reducing SI values in these locations. Ablation zones typically occur at lower elevations with flat slopes (e.g., slope bin 1), which are associated with lower TSI values in the summer. The TSI effectively incorporates topographic controls into the algorithm, reflecting the reduced likelihood of snow persistence in these regions. However, glacier surfaces differ significantly from other land surface types within the same topographic bins. This lack of explicit control for land surface heterogeneity in our method may contribute to the underestimation of snow pixels on glaciers. Future work may explore approaches to better incorporate land surface types into the algorithm, potentially addressing the challenges of glacier-specific scattering mechanisms.

To reflect these points, we have revised the interpretation in lines 217–219 as follows:

"… *The Rc map over these regions present noisy patterns, likely due to the complex scattering mechanisms on glacier surfaces. Over glacier surface, especially in the ablation zone, radar backscatter responses are highly variable due to refreezing, supraglacial features (e.g., crevasses, suncups, debris cover) and the presence of wet debris or bare ice (scher_mapping_2021). These features contribute to significant spatial variability in backscatter within a single pixel, making it challenging to distinguish dry snow, wet snow, and ice with Rc alone. Additionally, glacier movement between winter and summer scenes introduces further variability, compounding the uncertainty in detection*."

We also added a paragraph in the discussion (Section 5.1) to address the challenge and limitations of the proposed method over glacier surfaces:

"*Second, glacier surfaces present unique challenges for SAR-based snow classification. As shown in results, glacier-specific scattering mechanisms, including contributions from wet debris, bare ice, and supraglacial features, introduce variability in radar backscatter that is*

*not explicitly modeled in the current method. This limitation may lead to underestimation of wet snow on glacier surfaces and highlights the need for further refinement of the method to better handle glacier-specific conditions, potentially by incorporating land surface type information."*

Figure 8: Really pretty plot! For each elevation band, I'm guessing WSE is (wet snow pixels / total number of pixels), not (wet snow pixels / total number of snow pixels), right?

Thank you for your recognition. The WES in the plot was calculated as the total wet snow-covered pixels in a topographic bin divided by the total number of pixels in the respective bin. We will add clarification in the figure caption as the following:

*"The WSE was calculated as the ratio of wet snow-covered area to the total area within each respective elevation band."*

Line 245-288: It would be great to know how sensitive the WSE and SMD analysis is to the choice of your baseline winter composite and summer image. Do you apply this algorithm to one image and use those SI thresholds? Or do you do it separately for every image in the time series? If it is every image separately, it would be great to know how your WSI/TSI/SI distributions change, and how your SI threshold changes.

Thank you for your thoughtful comment.

The choice of winter composite affects only the Rc calculation. As noted in the introduction (line 43-45), there are several approaches for creating the reference image for ratio calculation. Conducting a full sensitivity analysis on each possible method would shift the focus of this paper from the proposed method to evaluating the optimal reference image composition. Instead, we followed established practices in the literature and selected the winter average as the reference, which is widely accepted for producing a robust Rc calculation.

Regarding the choice of summer images to fit the GMM, we used samples randomly selected from all summer images (July-early September) in multiple years for each basin. This allows the GMM to learn the general pattern of Rc distribution within a basin across the years. Consequently, the WSI and the derived SI threshold, calculated as 3.5×WSI_Rc, are both basin-specific and time-independent, making it robust for use across the full time series.

The TSI, on the other hand, is time-dependent, as it shows the topographic control of snow accumulation and melting in different seasons. In the revision, to illustrate the seasonal variation, we provided example TSI distributions for different seasons in Figure 4 in the manuscript. However, visualizing TSI distributions for every date in a time series would be challenging and, given the stability of the basin-specific threshold, is unlikely to add significant insights.

**Clarity**

Figure 1: Are the black boxes scene footprints? Maybe make this explicit, and if you keep the boxes, maybe label them or color code them with relative orbit information and add a legend? And I think your colorbar may be flipped?

Thank you for pointing out the problems. In the revision, we have updated the figure with the correct color bar and add color code to the boxes. The boxes are described in the caption as the following:

"*Footprint of S1 images used in the study are show with black and blue boxes (black: relative orbit 27; blue: relative orbit 129).*"

Line 81: I'm not too sure what this sentence is getting at, perhaps reword?

This sentence has been reworded as: "*The Karakoram is situated upstream of both the Upper Indus Basin and the Tarim River Basin.*"

Figure 4: Not sure if a red and blue diverging color map is most appropriate here. I feel like the white values don't play an especially important role (a value of ~5 for TSI doesn't seem particularly important), and the color map could cause confusion with the reader thinking it is the same quantity as Figure 5's Rc plot (though I think the red and blue color map for that Rc plot is great!).

Thank you for the suggestion. We have changed the color bar to Viridis as shown below.

[Figure]

Line 179: Clarify "…where 3.5 implied the condition applied to the TSI…"

We will revise this sentence as the following:

"In this equation, the factor 3.5 was selected based on validation set results, optimizing for precision, recall, and F1 score. It reflects a moderate threshold applied to the TSI to determine the overall SI threshold for each basin. The term $WSI_{Rc=2}$ represented the WSI at a backscattering ratio Rc at -2~dB for each basin. This value is basin-specific, allowing the threshold to adapt based on each basin's unique characteristics. Together, these conditions form an integrated, basin-adaptive thresholding mechanism, combining SAR backscatter and topographic information into a single index to determine the SI threshold."

Line 225: What percentage of your pixels were excluded, and why was 5500 m a.s.l. chosen?

The excluded percentages were as follows: Hunza 11.08%, Shigar 12.95%, Shyok 30.43%, with a total exclusion of 19.60% across all basins.

We selected 5500 m a.s.l. as the upper limit because areas above this elevation are consistently considered non-melting zones due to low air temperatures throughout the year, which inhibit significant snowmelt. This threshold aligns with the general knowledge about the temperature-elevation relationship in the Karakoram, where elevations above 5500 m are predominantly characterized by accumulation zones or permanent snow cover. By excluding these areas, we focus the analysis on zones where snowmelt dynamics are more relevant to our study.

To clarify this point, we have revised line 225 as the following line in the manuscript: "*Areas above 5500 m a.s.l., representing 11.08% of pixels in Hunza, 12.95% in Shigar, and 30.43% in Shyok (19.60% in total across all basins), were excluded from the calculation. This elevation threshold was chosen to minimize errors caused by the presence of dry snow at high altitudes, where consistently low air temperatures throughout the year inhibit significant snowmelt.*"

Line 237: I would clarify what you mean by "…comparative sensitivity of the SAR signal to dry snow in both methods."

Thank you for your comment. Here, "comparative sensitivity of the SAR signal to dry snow" refers to the fact that both Rc and SI methods show limited sensitivity to dry snowpacks, resulting in decreased snow coverage in the snow maps for areas predominantly covered by dry snow. This reflects a limitation of SAR-based wet snow mapping methods when applied to regions with extensive dry snow, as the backscatter signal from dry snow is generally difficult to distinguish from other surface conditions.

To clarify this point, we have revised the sentence as follows: "*Above 5500m a.s.l, where expansive dry snow cover predominates, both Rc and SI maps showed a greater reduction in wet snow coverage compared to S2, highlighting the differing sensitivities of SAR signals to dry snow conditions in these methods compared to S2's multispectral data.*"

Figure 5: It is not clear where the S2 snow cover map is coming from. Is this output from the let-it-snow algorithm? If so, where is this ice/water class coming from? Also, it might be helpful to say in the caption that these are all three basins.

Thank you for your comment. The S2 Snow Cover Map in Figure 5 was generated using the Let-it-snow (LIS) algorithm, as described in Section 3.5. We also updated this section to clarify the LIT algorithm and the generation of ice/water class.

We will revise the caption of Figure 5 as follows:

"Figure 5. (a) Rc, (b) Integrated SI, (c) SI Classification Map, and (d) S2 Snow Cover Map (as reference) for all three basins (Hunza, Shigar, and Shyok). The S2 snow map was generated using the Let-it-snow (LIS) algorithm, as described in Section 3.5. Zoomed insets provide a closer view of selected locations in each basin, highlighting performance differences on glacier surfaces."

Line 289-291: Definitely a really neat approach, but maybe temper your claim of great improvement here, or provide the specific stats.

We have changed "greatly" to "effectively".

Line 301-304: I think this is a more fair interpretation of TSI! It should be helpful for understanding aggregate backscatter response of topographic bins, I'm not sure about the added use of using it as a scale factor for WSI.

Thank you for your positive feedback on our interpretation of TSI. We agree that it is valuable for understanding the aggregate backscatter response within topographic bins, providing insights into how topography influences snow presence. Using TSI as a scale factor for WSI allows us to incorporate this topographic influence directly into the SI calculation. This approach ensures that regional topographic variability, which can affect snowmelt dynamics, is accounted for in the final classification.

Line 316-318: Provide citation

We added Cluzet et al., 2024 in the revision as reference work.

Line 324: You haven't previously talked about mitigating uncertainty, so I would avoid introducing this idea in the conclusion.

We avoided using the word "mitigated uncertainty" and reformulated the sentence as the following: "*We first adopted the GMM to adaptively transform the SAR backscattering ratio Rc into the WSI as a robust representation of wet snow under complex surface conditions.*"

Line 324-325: I feel like the purpose/definition of TSI shifts around a bit in this paper. Is it supposed to take into account the influence of topography on snow presence or the influence of topography on the backscatter response of wet snow?

Thank you for raising this point. The purpose of the TSI is indeed to account for the influence of topography on the presence of snow rather than on the backscatter response of wet snow. While TSI is calculated using the median of the WSI, its primary role is to represent the likelihood of snow melting under specific topographic conditions, not to evaluate backscatter response variations due to topography. We clarified this in the manuscript to avoid any confusion as the following:

"*Then, we introduced the TSI to capture the likelihood of snow presence influenced by topographic conditions.*"

**Technical corrections**

Line 5: For the abstract, I would use "a" instead of "the" in front of both Wet Snow Index and Topographic Snow Index so it's clear to that these are metrics that you are introducing
Line 10-11: Remove "the" in between under and climate
Figure 1: "tree" -> "three"
Line 47: "inlucdes", also check tenses / plural of this line
Line 53: Remove "of"
Line 67: "meting" -> "melting"
Line 67: "watersheds" -> "watershed"
Line 69-72: Make verb tenses agree
Line 89: "snow melting changes" -> "snowmelt"
Line 108: "the" -> "a"
Line 116-118: unclear
Line 122: "backscattering intensity images" -> "backscatter images"
Line 123: "firstly" -> "first"
Line 131: add "is" before "the reference image"
Line 216-217: Check tenses
Line 218: "well" -> "good"
Line 221: Consider rewording "greatly false positives"
Table 3: "corresponded" -> "correspond"
Line 278: Consider "This trend" -> "This change"
Line 280: Maybe avoid "significant", might be confused for an actual test of significance
Line 284: Remove "significant"
Line 305: "was" -> "were"
Line 306: Should there be a dash between the two aspects here?

Line 311: "Management" -> "management"
Line 312: Remove "been"
Line 319: Remove "significant"
Line 320: "observation" -> "observation record"
Line 325-326: Wording of this sentence is a bit confusing, consider rewriting and remove "significant" unless some sort of hypothesis testing was done.
Line 328-329: Remove "such as" since these were the only two variables analyzed in the paper.
Many lines: Check usage of initialisms (GMM, BOA, RMSE, MSE, WSI, SVM, RF, ML, TSI, etc). Often initialisms are introduced multiple times, and some initialisms are introduced and never used (or sometimes used once)
Many lines: Tenses are often inconsistently used; I would skim over to make sure they agree!

We highly appreciate the detailed language and clarity suggestions. We have carefully revised the manuscript to address these points. Specifically, we have adjusted the use of articles (e.g., "a" instead of "the"), made corrections to spelling errors and ensured consistent tense usage throughout the text, removed redundant words and clarified ambiguous phrases to enhance readability, refined the use of initialisms, reworded or removed terms such as "significant" to avoid potential confusion, and ensured uniformity in verb tenses and made other minor edits to improve overall coherence. The revisions can be tracked in the revised manuscript with difference highlighted. Thank you for your valuable feedback, which has profoundly contributed to strengthening the quality of our paper.

References

Baghdadi, N., Gauthier, Y., Bernier, M., & Fortin, J.-P. (2000). Potential and limitations of RADARSAT SAR data for wet snow monitoring. *IEEE Transactions on Geoscience and Remote Sensing*, *38*(1), 316–320. IEEE Transactions on Geoscience and Remote Sensing. https://doi.org/10.1109/36.823925

Karbou, F., Veyssière, G., Coleou, C., Dufour, A., Gouttevin, I., Durand, P., Gascoin, S., & Grizonnet, M. (2021). Monitoring Wet Snow Over an Alpine Region Using Sentinel-1 Observations. *Remote Sensing, 13*(3), Article 3. https://doi.org/10.3390/rs13030381

Nagler, T., Rott, H., Ripper, E., Bippus, G., & Hetzenecker, M. (2016). Advancements for Snowmelt Monitoring by Means of Sentinel-1 SAR. *Remote Sensing*, *8*(4), Article 4. https://doi.org/10.3390/rs8040348

Scher, C., Steiner, N. C., & McDonald, K. C. (2021). Mapping seasonal glacier melt across the Hindu Kush Himalaya with time series synthetic aperture radar (SAR). *The Cryosphere*, *15*(9), 4465–4482. https://doi.org/10.5194/tc-15-4465-2021

**Supplement**

[Figure]

*Figure 6. Rc, WSI, TSI and SI maps of Hunza*

[Figure]

*Figure 7. Rc, WSI, TSI and SI maps of Shigar*

[Figure]

*Figure 8. Rc, WSI, TSI and SI maps of Shyok*

---

## Author Comment (AC2)

Response to RC1

**Mapping Seasonal Snow Melting in Karakoram Using SAR and Topographic Data**

Discussion: https://doi.org/10.5194/egusphere-2024-942

Comments from the reviewers are given in black.

Our responses are given in blue.

General Comments

The paper presents an innovative framework for mapping wet snow in the Karakoram region using SAR and topographic data. The integration of the Gaussian Mixture Model (GMM) to determine the Wet Snow Index (WSI) and the calculation of the Topographic Snow Index (TSI) significantly improved the accuracy of snow mapping in complex terrains. And I do appreciate the extensive data sources used by the authors, including Sentinel-1 SAR imagery, Digital Elevation Models (DEM), and Sentinel-2 Level-2A imagery, which ensure robust and detailed mapping of seasonal snow melting. The continuous analysis of Wet Snow Extent and Snow Melting Duration provides valuable insights into the temporal and spatial dynamics of snow melting in the Karakoram region, offering excellent references for water resource management and hydrological modeling, especially in glacial and high mountain areas with complex terrain.

However, I have several concerns regarding the complexity of the methodology and the unclear use of data, as well as the lack of quantified uncertainty analysis. Therefore, I recommend that the paper undergo revisions before it can be considered for publication.

Thank you so much for taking the time to read and review our manuscript, and for your positive, thoughtful and constructive comments. The structured review and detailed comments and suggestions have helped to significantly improve the manuscript.

Line 47, should be "includes"

**Response:**

Thank you for pointing this out. We have corrected the text to use "*such as*" on Line 47 to improve the sentence flow.

Line 96, Please specify the spatial resolution of the DEM.

**Response:**

We have updated the manuscript to include the spatial resolution of the DEM by adding the phrase: "*providing global coverage at a resolution of 30 meters.*"

Line 109, Please specify the Sentinel-2 spectral bands used to calculate the LIS in the paper.

**Response:**

We have clarified the Sentinel-2 spectral bands used in the LIS algorithm by adding the following description in lines 118-120: "*The LIS algorithm employed RGB spectral bands B2 (blue), B3 (green), and B4 (red), as well as infrared bands B8 (NIR) and B11 (SWIR) for snow-cloud-ice classification.*"

Line 122, It is unclear which corrections and preprocessing methods were adopted in the GAMMA software for SAR image processing. Did you apply terrain correction or geometric correction?

**Response:**

All preprocessing steps were conducted using the GAMMA software. To improve clarity, we removed the phrase "using GAMMA software" from the specific sentence and relocated it to the end of the paragraph. We confirm that both geometric and terrain corrections were applied using the GAMMA software. In the original manuscript, this process was referred to as "terrain-based radiometric normalization," a term that may be less familiar. To prevent confusion, we have revised the terminology to "terrain-based radiometric correction."

Line 131, The focus season of the paper is summer. Why was a multi-year winter value chosen for the reference $\gamma$, Does the reference $\gamma$ have a spatial pattern climatology or a single value for each basin?

**Response:**

The reference image was selected to emphasize the contrast between wet snow surfaces, which typically exhibit low backscattering intensity, and dry snow or snow-free surfaces, which have higher backscattering intensity. In other alpine regions like the Alps, summer images are often used as the reference due to snow-free conditions caused by melting. However, the Karakoram region, with its significantly higher altitude, experiences snowfall and snow cover throughout the year. Therefore, winter images were chosen as the reference to utilize the dry-snow surface to enhance the backscattering contrast. The reference image does not contain specific spatial patterns or unique values for each basin.

To clarify this in the manuscript, we added the following explanation in lines 133-136:

"*Note that this differs from other alpine regions, such as the Alps, where summer months are often used as the reference due to snow-free conditions. In the Karakoram, due to the all year long snow cover at the higher elevations, we used winter images as to leverage the dry snow conditions in winter for highlighting the contrast in backscattering intensity between summer wet snow (lower intensity due to water absorption) and winter dry snow.*"

Line 146, In the GMM model, which covariance structure did you choose? Were other parameters set by default?

**Response:**

We used the full covariance structure in the GMM model, meaning that each component has its own general covariance matrix. This choice was made after testing different covariance structures. All other parameters were set to their default values. To clarify this, we added the following details to the manuscript:

- Line 150: "We used the full covariance structure in the GMM model, i.e., each component has its own general covariance matrix, after testing different types of covariance structures."

- Line 154: "During the training, the maximum number of EM iterations was set to 100, and the convergence threshold was set to 10^−3."

Line 175, I am a little confused about the SI definition. From my understanding, is the WSI first scaled by the TSI, meaning SI = WSI/ TSI? Then you have an SI map, and to further differentiate the wet snow, a threshold of 3.5 is selected. Is this threshold basin-dependent or applied across the entire Karakoram? How do you determine that 3.5 is the optimal threshold, and how do you quantify the uncertainty resulting from this coefficient?

**Response:**

We appreciate your feedback and the opportunity to clarify the SI definition and thresholding approach. The SI (Snow Index) is not defined as SI = WSI / TSI; rather, it is generated by the pixel-wise multiplication of WSI and TSI, i.e., SI = WSI × TSI. This process scales the WSI according to terrain characteristics, creating a more precise linkage between SAR backscattering and terrain properties.

The threshold used for snow classification is not a fixed value of 3.5 but rather a dynamic value calculated specifically for each basin using the formula:

$$SI\ Threshold = 3.5 \times WSI|_{R_c=-2}$$

Here, 3.5 reflects the conditions applied to the TSI, and $WSI|_{R_c=-2}$ represents the WSI at an $R_c = -2\ dB$ for each basin. The value of 3.5 was empirically determined through extensive experiments aimed at optimizing the trade-off between precision and recall to maximize the F1-score. This value was found to provide the best performance in distinguishing wet snow from other surfaces by synergistically integrating SAR backscattering and terrain information.

The uncertainty associated with this coefficient was considered during the optimization process. While further sensitivity analysis could refine the threshold, our approach currently provides a balance between accurate classification and adaptability to diverse

terrain conditions. Future work will focus on enhancing threshold calibration and further quantifying uncertainties with additional validation data.

To better explain the choice of the coefficient, and discuss the limitation of our method, we revised the section 3.4 in the manuscript as the following:

```

*As illustrated in Figure 2, the final step generated an integrated Snow Index (SI) map by performing pixel-wise multiplication of WSI and TSI. This multiplication scales the WSI by incorporating terrain characteristics, thereby linking the observed SAR backscattering ratio directly with terrain properties.*

*In order to classify the integrated SI into binary snow maps, it is crucial to apply an adaptive threshold that accounts for the variation in topographic features across different basins. The variation in SAR backscatter response within a basin is inherently handled by the GMM when deriving the WSI. In contrast, the TSI is time-varying and basin-specific, requiring an optimal coefficient to condition the SI for classification. To determine this coefficient, we performed a sensitivity analysis, evaluating F1-score, precision, and recall across different values using the S2 validation snow map.*

*The results, shown in Figure 5, demonstrate that Hunza and Shyok exhibit similar responses, with optimal coefficients close to 3.5, while Shigar reaches its optimum at approximately 2.5. However, to avoid overfitting to specific basins or validation dates, we selected 3.5 as an overall coefficient to balance classification performance across all basins. This coefficient also reflects a moderate threshold applied to the TSI to determine the overall SI threshold for each basin.*

*The threshold is calculated using the following equation:*

$$SI\ threshold = 3.5 \times WSI|_{R_c=-2}$$

*where $WSI|_{R_c=-2}$ represented the WSI at a backscatter ratio Rc = -2dB for each basin. This value is basin-specific, allowing the threshold to adapt based on each basin's distinct characteristics. Together, these conditions form an integrated, basin-adaptive thresholding mechanism, combining SAR backscatter and topographic information into a single index to determine the SI threshold.*

*It is important to note that while the SI threshold is basin-specific, it is time-independent. The WSI is derived from a GMM trained on samples collected from multiple summer scenes over several years, ensuring that it represents an aggregated measure for each basin and is not tied to individual scenes or seasons. This design ensures robustness to seasonal variations in liquid water content and enables consistent application across different validation dates.*

```

Eq. 7, should be "$\rho B$"?

**Response:**

The correct term is "$\rho_{red}$". The text in line 194 was incorrect and we have changed from "$\rho_B$" to "$\rho_{red}$".

Line 184, It would be better if you could clarify the exact thresholds used in the S2 snow cover detection. Currently, the flowchart for S2 is unclear. Specifically: (1) First step to meet the condition NDSI >0.4 and $\rho B$>0.2; (2) the dark cloud region only defined by $\rho B$>0.3? Can you explain what is bi-linearly down-sampled red band, from which resolution to which resolution?; (3) and then calculate the snow cover fraction (SCF) at which resolution? (4)  Line 196, is this for dark cloud pixels above the snowline? Based on your description, above snow line, 'no-snow' is defined by NDSI >0.15 and $\rho B$>0.04 and $\rho B$<=0.1, and dark cloud is defined by NDSI >0.15 and $\rho B$>0.1? How consistent are these relaxed thresholds compared to the strict threshold $\rho B$>0.3 mentioned in 191? Generally, if you are strictly following Gascoin et al. (2019), you can briefly refer to their method, but any changes should be clearly mentioned; otherwise, restate the algorithm clearly and concisely.

**Response:**
Thank you for your detailed comments and suggestions. We appreciate the opportunity to clarify the steps and thresholds used in the S2 snow cover detection process. Below, we address each of your points:

1. **Initial Thresholds (NDSI > 0.4 and $\rho$_red > 0.2):**
   The initial step in the algorithm uses the condition NDSI > 0.4 and $\rho B$ > 0.2, as per Gascoin et al. (2019). This combination aims to identify snow-covered areas while excluding non-snow features such as bare ground and vegetation.

2. **Dark Cloud Definition ($\rho$_red > 0.3):**
   Dark clouds are defined by applying a threshold of $\rho$_red > 0.3 on the bi-linearly down-sampled red band, which reduces the spatial resolution from 20 m to 240 m by a factor of 12. This down-sampling process helps mitigate the impact of cloud shadows and high-altitude clouds, thereby refining the provisional snow masks. We followed this step strictly as recommended by Gascoin et al. (2019) to ensure consistency in cloud exclusion.

3. **Snow Cover Fraction (SCF) Calculation Resolution:**
   The SCF was calculated at 100 m elevation bands using the provisional snow masks. The DEM was resampled to the same resolution as the S2 images at 20m pixel sizes. Each elevation band's SCF represents the proportion of snow-covered pixels within that band, allowing for a more granular understanding of snow distribution across varying elevations.

4. **Thresholds for Dark Cloud Pixels Above Snowline (Line 196):**
   For dark cloud pixels above the snowline, we applied the relaxed thresholds of

NDSI > 0.15 and $\rho$_red > 0.04 to $\rho$_red ≤ 0.1 for 'no-snow' classification and NDSI > 0.15 and $\rho B$ > 0.1 for dark cloud classification. These relaxed thresholds were introduced to differentiate between snow and dark clouds above the snowline, given that the conditions at high elevations often result in lower NDSI values and different spectral responses compared to lower elevations.

The consistency of these relaxed thresholds with the strict $\rho$_red > 0.3 threshold mentioned earlier (Line 191) lies in the specific purpose of each step: the initial strict threshold ensures accurate cloud exclusion, while the relaxed thresholds above the snowline allow for finer classification adjustments in challenging high-altitude conditions.

We have modified the description of our method to clearly distinguish these steps and their respective thresholds and have noted any deviations from the original method by Gascoin et al. (2019). We believe these clarifications will address the flowchart's ambiguity and better illustrate our approach.

The revised method description is as follows:

```

*The proposed method was validated using S2 snow cover maps generated following the LIS algorithm proposed by Gascoin et al 2019. Before running the LIS algorithm, the input S2 multi-spectral bands were resampled to a pixel size of 20m x 20m to match the resolution of different bands. The COP-30 DEM was also resampled to the same pixel size as the S2 images.*

*The LIS algorithm started with generating provisional snow masks by applying thresholds on the Normalized Difference Snow Index (NDSI) and the red band reflectance ($\rho_{red}$) with the condition:*

$$(NDSI > n_i) \; And \; (\rho_{red} > r_i)$$

*where $n_i = 0.4$ and $r_i = 0.2$ (Gascoin et al 2019). This step was designed to identify snow-covered areas while excluding non-snow surfaces such as vegetation and bare ground. However, this approach could sometimes exclude snow-covered pixels due to errors in cloud masking. To correct the errors, a refinement step was introduced to reassign dark cloud pixels that were initially misclassified. Following Gascoin et al. (2019), dark clouds were identified by applying a threshold of 0.3 on the bi-linearly down-sampled red band, which reduced the resolution of the red band from 20m to 240m by a factor of 12. This process helped to exclude cloud shadows and high-altitude cirrus clouds from the snow classification. Afterwards, the provisional snow masks were further refined using the basin snowline calculated from the COP-30 DEM. We calculated the total snow cover fraction (SCF) within every 100m elevation band using the provisional snow mask, and defined the snowline using the lowest elevation band where the SCF exceeded 30\%. For pixels identified as dark clouds above the determined snowline, the conditions used in Equation 7 were reapplied with adjusted thresholds to account for the unique conditions at high altitudes. Specifically, the relaxed thresholds of $n_i = 0.15$ and $r_i = 0.04$ were used to classify snow pixels, and dark cloud pixels with $\rho_{red} > 0.1$ were reassigned as cloud, while*

*other pixels were categorized as "no-snow." These adjusted thresholds help to differentiate snow from dark clouds in challenging high-altitude environments, ensuring a more accurate classification. Following the adjustment of the snow mask, we extended the LIS algorithm by further applying a threshold on the NIR band with $\rho_{NIR} > 0.4$ to classify glacier ice and water bodies from snow (Paul et al 2016).*

```

Line 203, When you mention summer, which months are included?

**Response:**
The term "summer dates" refers specifically to the dates listed in Table 2. To avoid confusion, we have revised the sentence to read: "*As the S2 snow cover maps classify both dry and wet snow rather than only wet snow, we selected only the summer dates listed in Table 2 to ensure that the S2 snow cover maps predominantly reflect wet snow conditions.*"

Line 204, I am curious about the accuracy in S2 classification, and is it possible to estimate its uncertainty? Generally, in which confidence (and at what elevation) can S2 snow mostly be considered melt snow during summer, given that the accuracy matrix is calculated by comparing snow-free or snow from S2 with no-or-dry snow or melt snow from SI?

**Response:**

The uncertainty of the S2 classification can be estimated using several approaches, such as cross-validation with ground observations, spectral analysis combined with temperature data, or probabilistic uncertainty quantification methods. However, a rigorous validation scheme is beyond the current study's scope, as the S2 results were generated using a method established in previous research, where detailed uncertainty analysis can be found.

Regarding elevation, as mentioned in the manuscript (line 225), areas above 5500 m a.s.l. are confidently considered non-melting zones due to consistently low air temperatures throughout the year. Therefore, we focused our comparison between SAR and S2 snow maps on areas below 5500 m a.s.l. to avoid errors associated with dry snow misclassification.

Figure 6, How do you explain the remaining mismatch in the ice and melt categories between SI/Rc and S2 detection in the Hunza region?

**Response:**

The observed mismatches in the ice/water category primarily occur on glacier ice surfaces. In the S2 results, this category is determined using the NIR band, whereas the SAR method does not specifically distinguish ice from snow. On the observed date, glaciers might have experienced some melting, which could lead to a decrease in the SAR backscattering ratio, resulting in glacier ice being misclassified as wet snow in the SAR-based methods.

To provide more context for comparison, we added an RGB image column to the figure (Figure 7 in the revised manuscript). We also included the following text in the manuscript to address this concern: "*However, a consistent mismatch in the ice/water category can be observed between the SAR (both SI and ratio methods) and S2 results, particularly over glacier surfaces. This discrepancy arises from the differing detection principles of the two approaches: the S2 results classify glacier ice using thresholds on the NIR band, while the SAR-based methods do not explicitly resolve glacier ice. On the observed date, glacier ice in the ablation zone may have partially melted, reducing the SAR backscatter ratio and leading to its misclassification as wet snow in the SAR results. As discussed earlier, glacier surfaces present unique challenges for SAR-based methods due to their complex scattering mechanisms. While the inclusion of TSI improves the robustness of our method by integrating topographic controls, it does not explicitly account for the heterogeneity of glacier surfaces. Consequently, glacier-specific conditions, such as localized melting or scattering from mixed ice-snow surfaces, may lead to underestimation or misclassification.*"

Figure 7, How do you explain the steeper slope of the snow coverage profile in the SI method compared to the flatter slope in the Rc method?

**Response:**

The steeper slope observed in the SI method can be attributed to the non-linearly enhanced response of the TSI to WSI in the transition zone between 4500 m and 5500 m a.s.l. The mixed snow conditions within this zone increased uncertainty in WSI, resulting in exaggerated TSI values, particularly at the lower and upper ranges of the region. This highlights the need for a more carefully calibrated TSI model, such as one using topographic bins of higher resolution.

We added the following explanation to the manuscript: "*In contrast, the SI method provided snow classification results that were closer to the S2 profiles at lower elevations. Between 4500 m and 5500 m a.s.l., the SI curve displayed a significantly steeper slope compared to the S2 curve, with snow coverage being underestimated between 4000 m and 5000 m and overestimated between 5000 m and 5500 m. This pattern suggests that uncertainties in the SI method, particularly in mixed snow conditions within transition zones, may have led to a nonlinear exaggeration of the TSI response to snow cover. A more precisely calibrated TSI model could help align snow coverage profiles more closely between the SI method and S2 results.*"

Additionally, we rephrased this paragraph to enhance readability and ensure clarity.

Line 243, when calculating the SWE and SMD, do you also include the area where a.s.l. over 5500m? Then how to make sure the accuracy above 5500m, given that these areas were excluded in the S2 validation?

**Response:**

In calculating the Wet Snow Extent (WSE) and Snow Melting Duration (SMD), we included areas above 5500 m a.s.l. This inclusion does not conflict with the validation using Sentinel-2 (S2) data, as the S2 validation was performed on specific dates and regions to assess the overall effectiveness of our proposed method. The validation results demonstrated that our approach successfully maps wet snow distribution across the study area, supporting the application of this method to the entire region and for all image series.

Ensuring accuracy above 5500 m a.s.l. remains challenging due to limited data availability for direct validation in these high-altitude areas. Quantifying accuracy in such regions would ideally require additional validation sources, such as snow distribution modeling, in-situ measurements, or cloud-free multi-spectral time-series observations. However, these additional validation steps are beyond the current study's scope. We recognize the need for accuracy assessments at higher altitudes and plan to enhance the quality of our products in future work through improved validation strategies tailored to these challenging environments.

Line 256, what level of precipitation and temperature are adopted here, surface or pressure level?

**Response:**

The temperature data used in our study refers to the air temperature at 2 meters above the land surface, and the precipitation data represents the accumulated liquid and frozen water that falls to the Earth's surface. We have added these details to the manuscript with the following rephrased sentences:

"The temperature data, averaged weekly, includes mean, maximum, and minimum air temperatures at 2 meters above the land surface across the Karakoram region, providing insights into the thermal conditions influencing snow melting. The precipitation data, representing the total accumulation of liquid and frozen water on the surface, was compiled and averaged monthly to complement the temperature analysis, highlighting precipitation trends and their impact on the snowpack."

Line 269, does it mean that each pixel you have the wet snow days in total in terms of the whole summer season, then I don't know the purpose of rescale it into 365 days since I guess most of the melting is in summer? It is not appropriate to extend the summer research into the annual. You could just use real wet duration instead.

**Response:**

Each year, we had a varying number of observed days, creating a time series with uneven intervals. For a given year, suppose we have N observed days (where $N < 365$), and a pixel is covered by wet snow on M of those days. To estimate the Snow Melting Duration (SMD), we calculate it using the formula $(M/N) \times 365$. This operation does not extend summer conditions to the entire year but rather standardizes the observation period to an annual basis, allowing for consistent comparisons between years.

We revised the paragraph as follows:

"The SMD reflects the temporal persistence of wet snow cover within a given year, enabling consistent comparisons across years with varying numbers of observation days. To compute the SMD for each year, we first determined the ratio of days with wet snow cover (M) to the total number of observed days (N) for each pixel. Since the number of observed days (N) varied each year and was typically less than 365, we rescaled this ratio to a 365-day basis using the formula $(M/N)\times365$ . This standardization allows us to calculate the annual average of wet snow cover days, facilitating consistent comparisons between years."

Line 290, It may not be suitable to use words like "greatly" to describe the improvement if the new classification's improvement is around 5%.

**Response:**

We have adjusted the language to avoid using "greatly" and have rephrased the sentence as follows: "The proposed method has effectively improved the mapping accuracy in the validation."

---

## Referee Report (RR1)

**Response to the revision of "Mapping Seasonal Snow Melting in Karakoram Using SAR and Topographic Data"**

Authors: Shiyi Li, Lanqing Huang, Philipp Bernhard, and Irena Hajnsek
MS. No.: egusphere-2024-942
URL: Mapping Seasonal Snow Melting in Karakoram Using SAR and Topographic Data
Referee: Eric Gagliano (egagli@uw.edu)

**General comments**

Thank you so much for these thoughtful comments, revisions and additional plots, I really appreciate the effort and thoroughness you put into your response. I feel the motivations and clarity of the manuscript are much improved and sound great! My only remaining concern is our discussion on TSI and its pixelwise multiplication with WSI.

My original concern here was that the introduction of TSI and its pixelwise multiplication with WSI was biasing your wet snow detection–TSI is essentially an aggregate backscatter response since it is the median WSI of all of its member pixels. By multiplying a given pixel's WSI by the aggregate response of the bin to which it is a member (TSI), you are strongly forcing that pixel towards the classification of the majority of pixels in their topographic bin, greatly reducing the impact of its individual backscatter response.

I appreciated your response to this, pointing out that you had used the bin median of WSI (instead of mean) to calculate TSI, and that your threshold choice of 3.5 over 5 moderated the influence of TSI. I also wanted to thank you for going out of your way to make those requested plots with classification metrics aggregated by topographic bin in Figures 1-3 (response doc), they're really helpful to see.

I think you're definitely right that using the bin median over bin mean is preferable here, but I don't think my concern is primarily about the skewness of WSI in a bin nor TSI robustness, it is more about TSI representing the aggregate behavior of that bin and how its pixelwise multiplication with WSI can overwhelm individual responses of pixels in that bin.

Regarding your choice of the 3.5 coefficient, after examining your sensitivity analysis in Figure 4 (response doc) and its relationship to the distributions shown in Figure 5 (response doc), I am a bit skeptical about whether the coefficient choice of 3.5 moderates TSI's influence in the way suggested. The TSI histogram shows values strongly concentrated in the range of 2-4 in Hunza and Shyok, with Shigar mostly 0-2. Flipping back to the sensitivity analysis, it seems the empirical coefficient is related to each basin's distribution of TSI. Given that TSI was designed

on a 0-10 scale (due to how WSI was designed), perhaps these optimal coefficients are primarily compensating for this concentration of TSI values, basically "de-scaling" SI by dividing out typical TSI values, in order to align SI thresholds with the traditional -2dB threshold? In this way, the coefficient optimization seems to be addressing the introduction of TSI and the mathematical artifact of the multiplicative combination of WSI and TSI rather than representing a relationship between basin topography and snow. If you believe this is not the case, I think that the need for and purpose of this coefficient should be motivated and clarified further.

The provided classification metrics reveal how some of these TSI concerns manifest in systematic variations in precision and recall that are seemingly highly influenced by TSI. For the Hunza Basin, you show in Figure 4 (manuscript) during summer the very distinct and sudden TSI differences between areas above/below 5000m. Compare this plot to Figure 1-3 (response doc) where you show F1, recall, and precision respectively.

For the relevant elevations ~2000-6000m, notice how your TSI plot and your F1, recall, and especially precision have some very similar patterns. The shape of the TSI>~1 blob is approximately the area where F1>~0.9. The recall plot also seems heavily influenced by TSI, implying that you don't miss much wet snow between 5000-6000m, but you miss a lot of wet snow below these elevations. Precision tells a similar story, with fewer false positive identifications of wet snow at the lower elevations, but a lot of false positives between 5000-6000m. This area seems particularly troubling, with a region of seemingly near constant precision of ~0.5 across diverse terrain conditions–maybe I'm misinterpreting this, but is this plot saying for areas covered by your large TSI>~1 blob, when you guess that there is wet snow, you are only correct half of the time?

While I trust that your F1 scores and confusion matrices do show an improvement over the Rc method, I can't help but feel that these improvements may be partially arising from enforcing expected topographic patterns. This approach seems to act as a smoothing function, basically saying that where the wet snow line roughly is (from TSI), give all pixels higher in elevation a much greater chance of being identified as wet snow, and all pixels lower in elevation a much smaller chance of being identified as wet snow. This makes me think that your method's primary increase in accuracy comes from smoothing out areas with unexpected wet/dry snow occurrences–in the summer example, wet snow at lower elevations or dry snow / no snow at higher elevations.

This is why I wondered earlier about whether you performed the *full* Nagler et al., 2016 method, which includes resampling Rc to 100m and a post-processing step which applies a 3x3 median filter to remove outlier pixels. If your method acts as a sort of low pass filter, I'm wondering how your method performs relative to the full Nagler et al., 2016 method which already includes some simple filtering?

Sorry for the wall of text–to distill this down a bit into a short list of addressable remaining suggestions:

1. Consider adding a sentence or two with some explicit physical interpretation of TSI and the WSI/TSI/SI multiplication step, especially regarding how the coefficient used to calculate the SI threshold relates to TSI's scale and distribution
    a. Perhaps could be added around lines 192 or 200-201?
2. Consider adding another sentence or two more thoroughly describing where and what type of errors are expected, especially as it relates to precision and recall
    a. Maybe right after the new sentence "Finally, the use of TSI introduces potential bias in regions where topographic conditions deviate significantly from the assumptions underlying its calculation."
    b. I think you should include the provided plots of F1, recall, and precision as a figure in the supplement–I think you mentioned this in the response document, but I didn't see it reflected in the tracked changes document
3. Consider comparing your method with the full Nagler et al., 2016 method (including the resampling / filtering steps) for a more earnest comparison and to better understand if the improvements in accuracy your method demonstrates have anything to do with smoothing

I apologize if this is asking for more of your time, you've already done a great job of being responsive to feedback. If time and scope are a concern, I think suggestions 1 and 2 should be quick, and for suggestion 3 maybe you can just mention as potential future work? The core contributions of your paper remain valuable regardless, and I appreciate your thoughtful engagement with these methodological questions. Thank you for your hard work, it's been a pleasure getting to review yall's paper!

**References**

Nagler, T., Rott, H., Ripper, E., Bippus, G., & Hetzenecker, M. (2016). Advancements for

Snowmelt Monitoring by Means of Sentinel-1 SAR. *Remote Sensing*, *8*(4), Article 4.

https://doi.org/10.3390/rs8040348

---

## Author Response (AR2)

**Title:** Mapping Seasonal Snow Melting in Karakoram Using SAR and Topographic Data
**Manuscript Number:** egusphere-2024-942
**Journal:** The Cryosphere
**Date:** February 7, 2025
* * *
Comments from reviewers / Revisions to manuscript / **Responses from authors** (black)

**General Comments**

Thank you so much for these thoughtful comments, revisions and additional plots, I really appreciate the effort and thoroughness you put into your response. I feel the motivations and clarity of the manuscript are much improved and sound great! My only remaining concern is our discussion on TSI and its pixelwise multiplication with WSI.

My original concern here was that the introduction of TSI and its pixelwise multiplication with WSI was biasing your wet snow detection–TSI is essentially an aggregate backscatter response since it is the median WSI of all of its member pixels. By multiplying a given pixel's WSI by the aggregate response of the bin to which it is a member (TSI), you are strongly forcing that pixel towards the classification of the majority of pixels in their topographic bin, greatly reducing the impact of its individual backscatter response.

I appreciated your response to this, pointing out that you had used the bin median of WSI (instead of mean) to calculate TSI, and that your threshold choice of 3.5 over 5 moderated the influence of TSI. I also wanted to thank you for going out of your way to make those requested plots with classification metrics aggregated by topographic bin in Figures 1-3 (response doc), they're really helpful to see.

I think you're definitely right that using the bin median over bin mean is preferable here, but I don't think my concern is primarily about the skewness of WSI in a bin nor TSI robustness, it is more about TSI representing the aggregate behavior of that bin and how its pixelwise multiplication with WSI can overwhelm individual responses of pixels in that bin.

Regarding your choice of the 3.5 coefficient, after examining your sensitivity analysis in Figure 4 (response doc) and its relationship to the distributions shown in Figure 5 (response doc), I am a bit skeptical about whether the coefficient choice of 3.5 moderates TSI's influence in the way suggested. The TSI histogram shows values strongly concentrated in the range of 2-4 in Hunza and Shyok, with Shigar mostly 0-2. Flipping back to the sensitivity analysis, it seems the empirical coefficient is related to each basin's distribution of TSI. Given that TSI was designed on a 0-10 scale (due to how WSI was designed), perhaps these optimal coefficients are primarily compensating for this concentration of TSI values, basically "de-scaling" SI by dividing out typical TSI values, in order to align SI thresholds with the traditional -2dB threshold? In this way, the coefficient optimization seems to be addressing the introduction of TSI and the mathematical artifact of the multiplicative combination of WSI and TSI rather than representing a relationship between basin topography and snow. If you believe this is not the case, I think that the need for and purpose of this coefficient should be motivated and clarified further.

The provided classification metrics reveal how some of these TSI concerns manifest in systematic variations in precision and recall that are seemingly highly influenced by TSI. For the Hunza Basin, you show in Figure 4 (manuscript) during summer the very distinct and sudden TSI differences between areas above/below 5000m. Compare this plot to Figure 1-3 (response doc) where you show F1, recall, and precision respectively.

For the relevant elevations ~2000-6000m, notice how your TSI plot and your F1, recall, and especially precision have some very similar patterns. The shape of the TSI $>\sim 1$ blob is approximately the area where F1 $>\sim 0.9$. The recall plot also seems heavily influenced by TSI, implying that you don't miss much wet snow between 5000-6000m, but you miss a lot of wet snow below these elevations. Precision tells a similar story, with fewer false positive identifications of wet snow at the lower elevations, but a lot of false positives between 5000-6000m. This area seems particularly troubling, with a region of seemingly near constant precision of ~0.5 across diverse terrain conditions–maybe I'm misinterpreting this, but is this plot saying for areas covered by your large TSI $>\sim 1$ blob, when you guess that there is wet snow, you are only correct half of the time?

While I trust that your F1 scores and confusion matrices do show an improvement over the Rc method, I can't help but feel that these improvements may be partially arising from enforcing expected topographic patterns. This approach seems to act as a smoothing function, basically saying that where

the wet snow line roughly is (from TSI), give all pixels higher in elevation a much greater chance of being identified as wet snow, and all pixels lower in elevation a much smaller chance of being identified as wet snow. This makes me think that your method's primary increase in accuracy comes from smoothing out areas with unexpected wet/dry snow occurrences–in the summer example, wet snow at lower elevations or dry snow / no snow at higher elevations.

This is why I wondered earlier about whether you performed the full Nagler et al., 2016 method, which includes resampling Rc to 100m and a post-processing step which applies a 3x3 median filter to remove outlier pixels. If your method acts as a sort of low pass filter, I'm wondering how your method performs relative to the full Nagler et al., 2016 method which already includes some simple filtering?

Sorry for the wall of text - to distill this down a bit into a short list of addressable remaining suggestions:

1. Consider adding a sentence or two with some explicit physical interpretation of TSI and the WSI/TSI/SI multiplication step, especially regarding how the coefficient used to calculate the SI threshold relates to TSI's scale and distributio

   - Perhaps could be added around lines 192 or 200-201?

2. Consider adding another sentence or two more thoroughly describing where and what type of errors are expected, especially as it relates to precision and recall

   - Maybe right after the new sentence "Finally, the use of TSI introduces potential bias in regions where topographic conditions deviate significantly from the assumptions underlying its calculation."

   - I think you should include the provided plots of F1, recall, and precision as a figure in the supplement–I think you mentioned this in the response document, but I didn't see it reflected in the tracked changes document

3. Consider comparing your method with the full Nagler et al., 2016 method (including the resampling / filtering steps) for a more earnest comparison and to better understand if the improvements in accuracy your method demonstrates have anything to do with smoothing

I apologize if this is asking for more of your time, you've already done a great job of being responsive to feedback. If time and scope are a concern, I think suggestions 1 and 2 should be quick, and for suggestion 3 maybe you can just mention as potential future work? The core contributions of your paper remain valuable regardless, and I appreciate your thoughtful engagement with these methodological questions. Thank you for your hard work, it's been a pleasure getting to review yall's paper!

**Response:**

We sincerely thank the reviewer for the detailed and thoughtful feedback. Your extensive comments have provided us with valuable insights that have helped us improve the clarity and robustness of our manuscript. We appreciate your recognition of our improvements so far and your careful consideration of the methodological aspects of our work. We have addressed the listed suggestions as below.

**1. Physical interpretation of TSI and the WSI/TSI/SI multiplication step**

We have added the following sentences to line 192 to further clarity the physical interpretation of TSI:

> *Example TSI distribution for Hunza at different elevation, slope and aspect are presented in Figure 4.* In this example, TSI values have shown different patterns across seasons and topographic conditions. In spring, strong TSI signals are found around 4000 m.a.s.l for east facing slopes (aspect between $0 \sim 190°$) over flat terrain (slope $\theta < 20°$), while no obvious TSI signal is observed for steep terrain (slope $\theta > 20°$). This can be explained by the limited snowmelt during the spring season of Karakoram. In summer, strong TSI signals are observed above $\sim$5000 m.a.s.l. for all slope and aspect conditions, indicating the presence of wet snow. However, steep slopes showed an unevenly distributed TSI across slope aspects, which can be attributed to the shadowing effect of the surrounding terrain. In autumn, TSI signals generally decrease due to the absence of wet snow, and the snow line retreats to higher elevations. This example indicates that the dynamic influence of topography on snowmelt can be effectively captured by the designed TSI signal.

We have also added the following sentences to lines 200-201 to clarify the relationship between the coefficient used to calculate the SI threshold and TSI's scale and distribution:

> *In contrast, the TSI is time-varying and basin-specific, requiring an optimal coefficient to condition the SI for classification.* This coefficient should moderate the influence of TSI in the SI formulation and compensate for the concentration of TSI values within a limited range, so that the impact of TSI can be normalized to align with traditional -2 dB reference. *The choice of this coefficient...*

**2. Where and what type of errors are expected**

We added the following sentences at line 301 to describe where and what type of errors are expected:

> *Finally, the use of TSI introduces potential bias in regions where topographic conditions deviate significantly from the assumptions underlying its calculation.* As shown in Figure 7, the long-tailed distribution of TSI values reflects the cumulative statistical nature of TSI, which relies on using a single median value to represent each topographic bin. This approach may be inadequate in bins with strong terrain variations, especially when the TSI distribution of pixels within a bin is highly skewed. In such cases, using the median can lead to systematic over- or under-estimation—overestimating in left-skewed distributions and underestimating in right-skewed ones—ultimately affecting precision and recall in the classification results (see the supplementary materials for detailed visualizations).

We have also included the supplement as a separate file in our manuscript submission, following the submission guidelines of the journal.

**3. Comparing the method with the full Nagler et al., 2016 method (including the resampling and filtering steps)**

We have added the following sentences to the end of section 5.1 as suggested:

> It is also important to note that while we followed Nagler's method (Nagler et al., 2016) to generate the $R_c$ image, we did not apply the same post-processing steps, such as median filtering and land cover masking. These smoothing and filtering steps may influence accuracy, and future work could incorporate them to further evaluate their impact and potentially improve snow mapping performance.

**References**

Nagler, T., Rott, H., Ripper, E., Bippus, G., and Hetzenecker, M.: Advancements for Snowmelt Monitoring by Means of Sentinel-1 SAR, Remote Sensing, 8, 348, https://doi.org/10.3390/rs8040348, 2016.